# 3DPoV: Improving 3D understanding via Patch Ordering on Videos

## Abstract

Visual foundation models have achieved remarkable progress in scale and versatility, yet understanding the 3D world remains a fundamental challenge. While 2D images contain cues about 3D structure that humans readily interpret, deep models often fail to exploit them, underperforming on tasks such as multiview semantic consistency–crucial for applications including robotics and autonomous driving. We propose a self-supervised approach to enhance the 3D understanding of vision foundation models by (i) introducing a temporal nearest-neighbor consistency loss that finds corresponding points across video frames and enforces consistency between their nearest neighbors, (ii) incorporating reference-guided ordering that requires patch-level features to be not only expressive but also consistently aligned, and (iii) constructing a mixture of video datasets tailored to these objectives, thereby leveraging rich 3D information. Our method, 3DPoV, achieves state-of-the-art performance in keypoint matching under viewpoint variation, as well as in depth and surface normal estimation, and consistently improves a diverse set of backbones, including DINOv3.

## 1 Introduction

Recent advances in dense self-supervised learning have yielded feature representations that are remarkably effective for a variety of vision tasks, including object part recognition, dense retrieval, and semantic matching. Models like DINO (Caron et al., 2021) and its successors demonstrate that fine-grained correspondence can emerge even without explicit labels. However, a critical shortcoming remains: robustness to viewpoint change. When the camera pose shifts, these representations often degrade substantially, revealing a lack of true 3D spatial understanding.

This challenge is especially important in real-world scenarios where objects are seen from multiple perspectives, and consistent recognition across views is crucial. Existing self-supervised approaches based on static images or temporally adjacent frames–while effective in learning texture and semantics–struggle to capture geometric cues like depth, structure, or object permanence under motion. This gap has been increasingly highlighted by benchmarks like Probe3D (El Banani et al., 2024), which systematically exposes these limitations across keypoint matching, depth prediction, and surface normal estimation tasks.

To address this, we propose **3DPoV (3D understanding via Patch Ordering on Videos)**, a post-training strategy for enhancing multiview spatial consistency by enforcing temporal alignment across tracked patches. Our method builds on the insight that viewpoint changes induce systematic deformations in patch-level similarity patterns. By supervising the relative ranking of features extracted along point tracks over time, 3DPoV encourages the network to learn descriptors that remain consistent across large temporal and viewpoint shifts.

Unlike prior approaches such as TimeTuning (Salehi et al., 2023) and MoSiC (Salehi et al., 2024), which operate through temporal propagation of segmentation maps, or NeCo (Pariza et al., 2025), which focuses on intra-image part ordering, our framework directly aligns patch-wise relationships across frames. It leverages differentiable sorting (Petersen et al., 2022) to compare similarity structures over reference patches, and uses a teacher-student setup grounded in explicit temporal tracking to provide stable supervision under motion and occlusion.

By leveraging video sequences and lightweight fine-tuning, 3DPoV instills emergent 3D reasoning, with consistent gains across all Probe3D tasks–particularly under large viewpoint changes, occlusion, and lighting variation. Our approach narrows the gap between 2D feature learning and 3D understanding, offering an efficient and scalable path to enhance foundation models for geometry-aware visual reasoning. The main contributions of 3DPoV are as follows:

- We introduce a temporal permutation loss anchored by point tracks, which supervise the relative ordering of patch features across frames. This directly trains the model to produce viewpoint-invariant descriptors without relying on crops or masks.

- We propose a teacher–student setup where reference frames are also passed through the student–unlike prior works–yielding features that are both discriminative and sortable under motion and occlusion; stability is further ensured through a reference pool mixing external frames with internal samples from the same video

- We demonstrate that 3DPoV achieves consistent improvements across all Probe3D difficulty regimes. Unlike prior approaches that trade robustness at large viewpoint shifts for small-viewpoint gains, our method improves uniformly across viewpoint variation, occlusion, and lighting changes.

## 2 BACKGROUND

**Self-supervised learning on videos** has leveraged temporal coherence to improve semantic consistency, but often without explicitly modeling spatial alignment. TimeTuning (Salehi et al., 2023) propagates cluster assignments across frames to stabilize semantics, while MoSiC (Salehi et al., 2024) strengthens this with point tracks for improved consistency. However, both methods remain centered on propagating semantic groupings rather than directly optimizing for viewpoint-robust spatial understanding.

**Spatially-aware ordering methods** such as NeCo (Pariza et al., 2025) address viewpoint sensitivity in images by supervising the relative ordering of patch similarities via differentiable sorting. This approach enhances local spatial structure and yields more context-aware representations, making it particularly relevant to our work. However, NeCo is restricted to static images and overlapping crops, which assume a fixed viewpoint and discard the global context that intrinsically encodes spatial structure. These assumptions limit its applicability to videos, where motion and viewpoint changes dominate.

Evaluation frameworks such as Probe3D (El Banani et al., 2024) expose these gaps by probing robustness under viewpoint changes across tasks like keypoint matching, depth estimation, and surface normal prediction. Existing models tend to perform well under small viewpoint differences but suffer a sharp drop in accuracy as the viewpoint gap increases, highlighting the need for methods that improve consistently across all regimes. *Leveraging multiview supervision has recently emerged as a promising direction for enhancing 3D correspondence (You et al., 2024; Ruan et al., 2024)* More recently, models such as DINOv3 (Siméoni et al., 2025) have explicitly targeted these evaluations, reporting strong results and emphasizing the growing role of geometry-aware benchmarks in guiding self-supervised learning. In addition, Probe3D combines quantitative metrics with qualitative inspection, offering a diagnostic lens into whether models truly encode intrinsic 3D structure rather than relying on priors, appearance, or texture cues. The systematic gaps highlighted by Probe3D motivate our approach, which is designed to improve spatial consistency across viewpoint variation.

## 3 METHOD

We propose **3DPoV**, a frameworl for learning temporally consistent dense features from videos by leveraging point tracks and patch-level ordering. The method builds on a teacher–student architecture, where the student processes video frames independently and the teacher provides a stable anchor frame for supervision (Figure 1). To enforce temporal consistency, we track a grid of points across frames and extract features at aligned locations.

Rather than matching features directly, we align the *relative similarity structure* of tracked patches over time. For each frame, we compute similarity rankings with respect to a shared set of reference features and use differentiable sorting to obtain soft permutation matrices. The student is then trained

to match the teacher's anchor-frame permutations, encouraging viewpoint-invariant descriptors that remain consistent under motion, occlusion, and appearance changes.

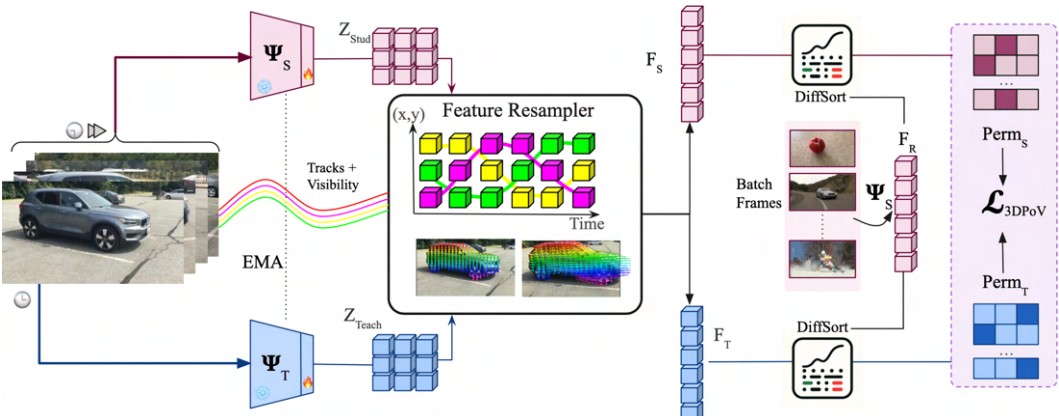

Figure 1: **3DPoV**: Learning 3D-aware representations via Patch Ordering in Videos. We begin by extracting motion trajectories $\text{Traj}_{f,i}$ from raw video clips using CoTrackerV3. The video is parsed into frames, and each is processed by the student and teacher networks to produce feature maps $Z^{\text{stu}}, Z^{\text{teach}} \in \mathbb{R}^{p \times d \times nf}$. Using the tracked coordinates from $\text{Traj}_{f,i}$, we resample features to obtain patch sequences $F_s$ (student) and $F_t$ (teacher). Reference features $F_r$ are extracted from other batch frames using the student network. Pairwise cosine distances $D_{i,j}$ are computed between $F_s$ and $F_r$, and between $F_t$ and $F_r$. These distances are sorted via a Differentiable Sorting module, producing permutation matrices $\text{Perm} \in \mathbb{R}^{np_q \times np_r \times np_r}$ that enforce consistent patch ordering across time.

**Preliminaries**   Given a video clip $X \in \mathbb{R}^{h \times w \times c \times nf}$, where $h \times w$ is the spatial resolution, $c$ the number of channels, and $nf$ the number of frames, we extract dense patch-level features using a Vision Transformer (ViT) (Dosovitskiy et al., 2021) backbone. Each frame is encoded independently by a student network $\Psi_S$, while the first frame is also processed by a teacher network $\Psi_T$, updated as an exponential moving average (EMA) of the student.

**Point tracking across frames**   To obtain spatial correspondences over time, we leverage an off-the-shelf point tracking module to estimate the trajectories $\text{Traj}_{f,i}$ and visibility masks $\text{Vis}_{f,i}$ for a set of points initialized on the first frame. Trackers such as CoTrackerV3 (Karaev et al., 2024) are capable of producing temporally consistent tracks over long video sequences, while being robust to challenges such as occlusion, lighting variation, and large viewpoint changes (Figure 2).

Specifically, we initialize a regular grid of size $g \times g$ on the first frame, yielding $N = g^2$ points with coordinates $(x_i, y_i)\}_{i=1}^N$. Given the video clip $X$ and this grid, the tracker predicts the trajectories of all $N$ points across the sequence as:

$$\text{Traj}_{f,i} := \texttt{Tracker}(X, (x_i, y_i)) \in \mathbb{R}^{nf \times N \times 2}, \tag{1}$$

where $\text{Traj}_{f,i}$ denotes the coordinate location of the $i^{\text{th}}$ point in each frame $f$, for all $nf$ frames in the video.

Since tracking is initialized on the first frame, all points are guaranteed to be visible at $t = 0$. We therefore designate frame 0 as the anchor and extract its features with the teacher network, which provides a stable reference throughout training. Later frames, processed by the student, may contain occlusions or appearance changes; aligning them with the clean anchor frame encourages viewpoint- and occlusion-invariant representations.

**Feature Extraction and Alignment**   Features $Z^{\text{stu}}, Z^{\text{teach}} \in \mathbb{R}^{p \times d \times nf}$ are extracted from raw frames using the student $\Psi_S$ and teacher $\Psi_T$ networks, where $p$ denotes the number of patches, $d$ is the feature dimension, and $nf$ the number of frames per video. While NeCo (Pariza et al., 2025) leverages ROI align to extract overlapping patches between paired crops of a single image, our approach instead samples full-frame patches and uses point tracks to extract aligned patch trajectories

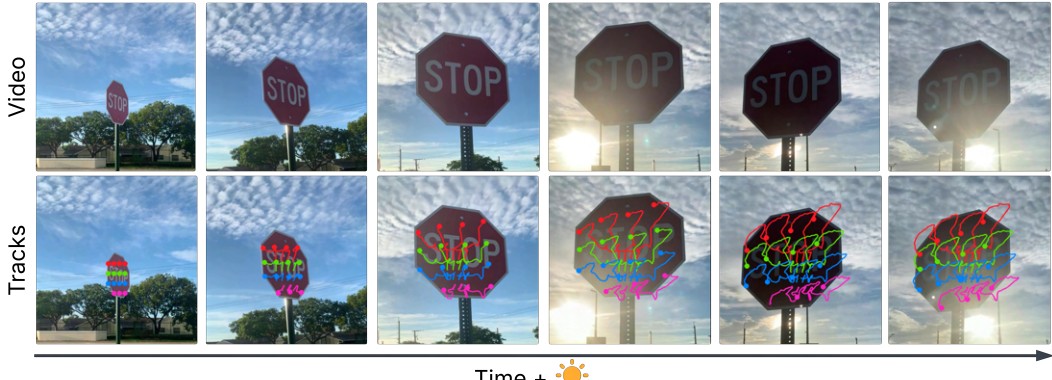

Figure 2: CoTrackerV3 maintains high tracking accuracy across lighting changes, viewpoint shifts, and forward camera motion in a CO3D sample.

throughout time. Given a tracked trajectory $\text{Traj}_{f,i}$ for the $i^{\text{th}}$ point, we sample back corresponding patch features from $Z^{\text{stu}}$ and $Z^{\text{teach}}$ to obtain temporally aligned patch sequences $F_s$ and $F_t$.

To balance generalization and alignment quality, we consider two strategies for retrieving patch features from tracked coordinates. In resized sampling, feature maps are upsampled to the input resolution and features are retrieved via nearest-neighbor indexing. In latent-space sampling, features are extracted directly from the native feature grid using bilinear interpolation.

**Similarity via Differentiable Sorting**  To supervise the temporal consistency of patch features, we adopt a differentiable sorting mechanism that aligns the relative similarity structure of patches over time. Rather than enforcing direct feature similarity between frames, we compare the *ranking distributions* of each patch with respect to a shared set of reference features. This encourages the model to learn a structured, viewpoint-invariant representation of similarity–crucial for robust dense correspondence.

For each video clip, we construct a reference feature bank by sampling local crops from frames of other videos in the batch, as well as from a non-anchor frame $k \neq 0$ of the current video. These reference crops preserve spatial layout and introduce both intra-video and inter-video diversity. Instead of cropping raw input images, we apply spatial cropping in feature space after forwarding the reference frames through the student network $\Psi_S$. This yields a reference feature bank of patch features $F_r \in \mathbb{R}^{B \times np_r \times d}$, where $np_r$ is the number of reference patches per sample and $d$ is the feature dimension.

As such, the Differentiable Sorting module operates on a per-sample basis. It receives a set of query patch features $F_q \in \mathbb{R}^{B \times np_q \times d}$, extracted from either the student at a future frame $t > 0$ or the teacher at the anchor frame $t = 0$, where $np_q$ denotes the number of query patches. It also receives a set of reference features $F_r \in \mathbb{R}^{B \times np_r \times d}$ obtained from the reference bank.

To compare the query features with the reference features, we compute cosine similarity:

$$S_{i,j} = \frac{\langle F_q^i, F_r^j \rangle}{\|F_q^i\| \cdot \|F_r^j\|}, \quad D_{i,j} = 1 - S_{i,j} \tag{2}$$

for $i \in [1, np_q], j \in [1, np_r]$. Each row of $S$ encodes the similarity between one query patch and all reference patches. Since our goal is to capture relative ordering rather than absolute scores, we pass the distance matrix $D = 1 - S$ to the differentiable sorting module (Petersen et al., 2022) which outputs soft permutation matrices $P$ that approximate the ranking distribution of each query patch over the reference set. Full details of the sorting procedure are provided in Appendix B.

**Patch-Wise Permutation Loss for Temporal Alignment**  To supervise the temporal consistency of patch-level features, we compare the sorting behavior of the student network across time to that of the teacher network at a fixed anchor frame. Rather than enforcing direct similarity in feature space,

we align their respective soft permutation matrices over a shared set of reference patches. This encourages the student to match the teacher's viewpoint-invariant similarity structure, even under occlusions and appearance shifts.

Let $F_t^S \in \mathbb{R}^{B \times np_q \times d}$ denote student features at a future frame $t > 0$, and $F_0^T$ the teacher features at the anchor frame $t = 0$. For each of the $N_{\text{ref}}$ reference crops $F_r^R \in \mathbb{R}^{B \times np_r \times d}$, we compute soft permutation matrices via differentiable sorting:

$$P_{t,r}^S = \text{DiffSort}(F_t^S, F_r^R) \qquad\qquad P_{0,r}^T = \text{DiffSort}(F_0^T, F_r^R) \qquad (3)$$

Each soft permutation matrix $P \in \mathbb{R}^{B \times np_q \times np_r \times np_r}$ encodes, for every query patch, a distribution over the ranked positions of reference patches.

**Patch-wise Cross-Entropy Loss** We supervise the student permutation matrix $P_{t,r}^S$ with respect to the teacher matrix $P_{0,r}^T$. For each query patch $i$, we compute the cross-entropy where the student distribution provides the weighting:

$$\mathcal{L}_{\text{CE}}^i = -\sum_{j=1}^{np_r} P_{0,r}^S[i,j] \cdot \log\left(P_{t,r}^T[i,j] + \epsilon\right) \qquad (4)$$

This formulation encourages the student to place probability mass in regions where the teacher also provides support, while simultaneously promoting disentangled and confident predictions. In practice, this leads to sharper spatial rankings and improves patch-level discrimination. We then average this loss across all query patches $i$ and samples $b$ in the batch:

$$\mathcal{L}_{\text{CE}}^{(t,r)} = \frac{1}{B} \sum_{b=1}^{B} \frac{1}{np_q} \sum_{i=1}^{np_q} \mathcal{L}_{\text{CE}}^{(b,i)} \qquad (5)$$

**Visibility-Weighted Loss** To account for occlusion and tracking failures, we weight each patch by its visibility at both the anchor and current frames. Let $v_{0,t}^{(b,i)} = V_{0,i}^{(b)} \cdot V_{t,i}^{(b)}$ denote the joint visibility of patch $i$ in sample $b$.

The visibility-weighted cross-entropy becomes:

$$\mathcal{L}_{\text{CE}}^{(t,r)} = \frac{1}{B} \sum_{b=1}^{B} \sum_{i=1}^{np_q} \frac{v_{0,t}^{(b,i)}}{\sum_j v_{0,t}^{(b,j)} + \epsilon} \cdot \mathcal{L}_{\text{CE}}^{(b,i)} \qquad (6)$$

**Final Loss Across Time and References** To enforce alignment throughout the sequence, we apply the patch-wise loss across all compared frames $nf - 1$ and all references $r = 1, \ldots, N_{\text{ref}}$. The final permutation alignment loss is:

$$\mathcal{L}_{\text{3DPoV}} = \frac{1}{nf - 1} \sum_{t=t_{\text{start}}}^{T} \frac{1}{N_{\text{ref}}} \sum_{r=1}^{N_{\text{ref}}} \mathcal{L}_{\text{CE}}^{(t,r)} \qquad (7)$$

This objective encourages the student network to produce temporally aligned, viewpoint-consistent patch-level rankings relative to shared reference crops (Figure 5)–anchored by the teacher signal– while softening the contribution of low-confidence or occluded regions via visibility weighting. *Further details on loss formulation and design decisions are provided in Appendix C.*

An equally important factor is the choice of fine-tuning data, which plays a central role in shaping the model's ability to learn viewpoint-invariant and geometry-aware representations from videos. To capture complementary aspects of variability, we fine-tune on a blend of three datasets: (i) CO3D (Reizenstein et al., 2021), which provides long object-centric multiview sequences with large viewpoint shifts; (ii) DL3DV (Ling et al., 2024), which offers diverse dynamic scenes and spatial layouts; and (iii) YouTube-VOS (Xu et al., 2018), which introduces unconstrained motion, occlusions, and real-world camera trajectories. Together, this mixture spans single-object, scene-level, and natural video variability, supporting robust learning of dense, temporally consistent features. Full dataset descriptions and preprocessing details are provided in Appendix D.

## 4 EXPERIMENTS

We evaluate our method on the Probe3D benchmark (El Banani et al., 2024), which assesses 3D spatial understanding through keypoint matching, depth estimation, and surface normal estimation. *We fine-tune the last two Transformer blocks (blocks 10 and 11) while keeping the rest of the network frozen.* Unless otherwise stated, we use the DINOv2-R backbone. All comparisons are made against models with identical backbone architectures, isolating the effect of our method. To ensure fair placement of results, we reproduce all Dino baselines from (El Banani et al., 2024) and use publicly available checkpoints for prior post-training baselines (TimeTuning, NeCo, MoSiC). Dataset details and reproduction studies are provided in Appendix E and Appendix H.

**Keypoint Matching.** We evaluate on SPair-71k (2D human-annotated keypoints) and Navi (synthetic data with 3D geometry and calibrated cameras). On SPair, recall is measured by predicting target keypoints from feature similarity. Results are reported across viewpoint bins (small, medium, large) as well as the "All" split, which aggregates all pairs but is biased toward small-viewpoint cases. Navi supports 3D-aware evaluation: correspondences are matched directly in 3D and assessed both by Euclidean error in 3D space and reprojection error in 2D. We report recall at multiple thresholds and analyze results as a function of relative camera rotation. Full experimental details are deferred to Appendix E.

Table 1 reports results on SPair-71k. NeCo improves over its baseline mainly for small viewpoint differences but degrades sharply under larger shifts. In contrast, 3DPoV surpasses both DINO baselines and NeCo while maintaining balanced performance across all viewpoint bins, demonstrating stronger spatial consistency under diverse transformations. Segmentation-focused approaches such as TimeTuning and MoSiC achieve temporal semantic propagation but fail to retain the spatial discrimination required for robust keypoint matching. This indicates that improvements in semantic consistency over time do not directly translate into stronger spatial semantic correspondence.

| Model | *Backbone* | Data | S / 0 | M / 1 | L /2 | All |
|---|---|---|---|---|---|---|
| DINO | ViT-S/16 | IN-1k | **28.34** | **23.38** | **24.44** | **25.63** |
| TimeTuning | DINOv1-S/16 | YTVoS | 26.76 | 22.48 | 23.45 | 23.96 |
| MoSiC | DINOv1-S/16 | YTVoS | 26.73 | 21.97 | 22.98 | 23.76 |
| DINO | ViT-B/16 | IN-1k | 30.19 | 24.22 | 24.35 | 26.39 |
| NeCo | DINOv1-B/16 | COCO | 30.24 | 24.45 | 23.10 | 26.32 |
| 3DPoV | DINOv1-B/16 | CO3-YT-DL | **31.77** | **25.74** | **25.80** | **28.16** |
| DinoV2R | ViT-B/14 | LVD | 58.20 | 51.56 | 53.41 | 53.47 |
| NeCo | DINOv2R-B/14 | COCO | 59.57 | 49.06 | 52.35 | 54.42 |
| MoSiC | DINOv2R-B/14 | YTVoS | 56.37 | 50.70 | 51.75 | 51.72 |
| *3DCorrEnhance* | DINOv2R-B/14 | - | 59.61 | 52.16 | 54.39 | 54.64 |
| 3DPoV | DINOv2R-B/14 | CO3-YT-DL | **60.16** | **52.79** | **54.50** | **55.40** |
| DinoV3 | ViT-B/16 | LVD | 61.95 | **48.69** | 46.77 | 55.73 |
| 3DPoV | DINOv3-B/16 | CO3-YT-DL | **62.24** | 48.56 | **46.81** | **55.84** |

Table 1: SPair-71k viewpoint difference. 0: No significant view difference (same view or minimal changes), 1: Moderate viewpoint difference, 2: Large viewpoint difference. DINOv2R: DINOv2 with registers

A similar pattern is observed on Navi (Table 2). While NeCo shows gains on SPair, its improvements do not transfer as effectively, reflecting the added difficulty of enforcing 3D-consistent correspondences. 3DPoV, on the other hand, consistently improves across all relative viewpoint bins, with only a minor drop in the $\theta_{60}^{180}$ range for the DINOv2 variant, underscoring its robustness under large viewpoint changes.

Finally, the breakdown in Table 6a and Table 6b shows that 3DPoV achieves consistent gains in both 3D correspondence accuracy and 2D reprojection alignment across all thresholds. This dual improvement highlights that the learned features are geometrically faithful in 3D space while also preserving accurate alignment in 2D.

**Depth and Surface normal estimation** We evaluate our model's geometric understanding using depth and surface normal estimation on the Navi benchmark, following the standardized protocol

| Model | *Backbone* | Data | $\theta_0^{15}$ | $\theta_{15}^{30}$ | $\theta_{30}^{60}$ | $\theta_{60}^{180}$ |
|---|---|---|---|---|---|---|
| DINO | ViT-S/16 | IN-1k | **84.36** | 55.17 | **34.58** | **20.48** |
| TimeTuning | DINOv1-S/16 | YTVoS | 80.81 | 52.61 | 33.93 | 19.96 |
| MoSiC | DINOv1-S/16 | YTVoS | 80.21 | 52.07 | 33.37 | 19.59 |
| DINO | ViT-B/16 | IN-1k | 86.13 | 56.92 | 33.37 | 19.74 |
| NeCo | DINOv1-B/16 | COCO | 84.94 | 53.52 | 31.80 | 18.47 |
| 3DPoV | DINOv1-B/16 | CO3-YT-DL | **86.42** | **57.18** | **33.77** | **20.42** |
| DinoV2R | ViT-B/14 | LVD | 87.92 | 67.74 | 47.18 | **31.57** |
| NeCo | DINOv2R-B/14 | COCO | 88.69 | 64.61 | 43.47 | 28.68 |
| MoSiC | DINOv2R-B/14 | YTVoS | 87.11 | 66.49 | 46.85 | 31.55 |
| *3DCorrenhance* | DINOv2R-B/14 | - | 87.92 | 67.74 | 47.18 | **31.57** |
| 3DPoV | DINOv2R-B/14 | CO3-YT-DL | **89.22** | **69.23** | **47.48** | 31.33 |
| DinoV3 | ViT-B/16 | LVD | 94.40 | 74.73 | 48.64 | **31.45** |
| 3DPoV | DINOv3-B/16 | CO3-YT-DL | **94.47** | **74.74** | **48.65** | 31.36 |

Table 2: Navi Performance Comparison Across Models with performance binned for different relative viewpoint changes between image pairs. Best results are in bold. DINOv2R: DINOv2 with registers

introduced in Probe3D. This evaluation tests whether the learned features encode meaningful 3D spatial geometry beyond keypoint-level correspondences.

Since the backbone models do not inherently predict depth or surface normals, we follow the Probe3D protocol on training lightweight linear probes on top of frozen features for each task. This setup isolates the quality of the learned representations, ensuring that performance reflects spatial awareness embedded in the features rather than downstream training capacity.

In line with (El Banani et al., 2024), we conduct both quantitative evaluation using ground-truth 3D signals and qualitative inspection to better interpret the spatial reasoning captured by the features. Full definitions of the evaluation metrics are deferred to Appendix E.

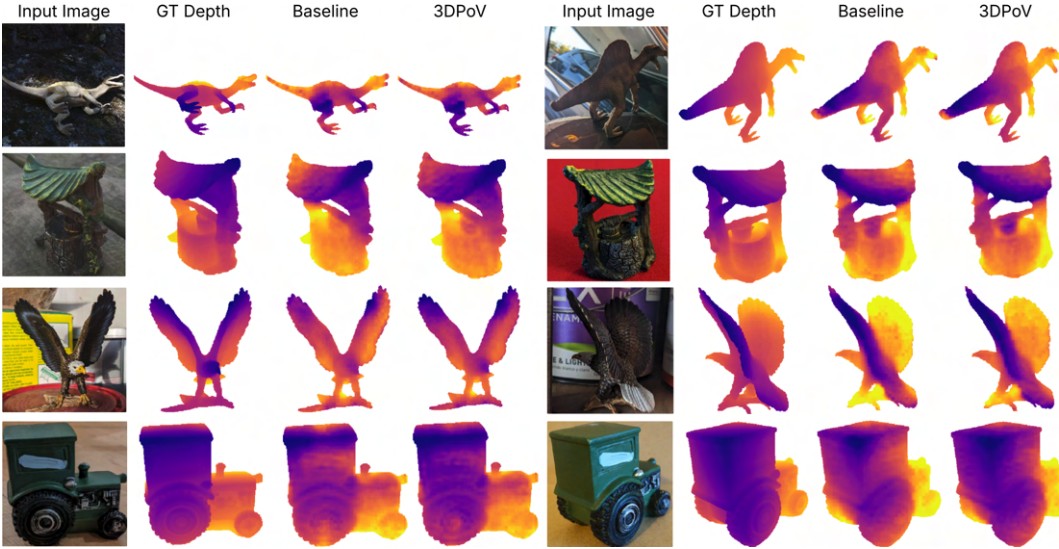

Figure 3: **Depth Qualitative Examples.** Comparing predicted depth maps from Baseline (DinoV2-reg) and **3DPoV**. Ground truth (GT) depth is provided for reference

Qualitative depth results (Figure 3) show that 3DPoV produces more coherent maps than the baseline, preserving boundaries and geometric detail across diverse object types. For instance, in the dinosaur example, our method resolves the lower leg despite heavy shadow, and on the tractor it avoids interpreting a painted stroke as spurious geometry, yielding a more plausible depth map. These improvements align with the quantitative gains in Table 3.

| Model | Backbone | Scale-Aware | | | | Scale-Invariant | | | |
|---|---|---|---|---|---|---|---|---|---|
| | | $\delta_1 \uparrow$ | $\delta_2 \uparrow$ | $\delta_3 \uparrow$ | RMSE $\downarrow$ | $\delta_1 \uparrow$ | $\delta_2 \uparrow$ | $\delta_3 \uparrow$ | RMSE $\downarrow$ |
| DINO | ViT-B/16 | 47.16 | 73.83 | 86.70 | 0.1237 | 58.64 | 81.85 | 90.43 | 0.1022 |
| 3DPoV | DINOv1-B/16 | **47.93** | **74.77** | **87.45** | **0.1218** | **58.83** | **82.20** | **90.67** | **0.1014** |
| DinoV2R | ViT-B/14 | 57.62 | 82.49 | 91.97 | 0.0960 | 68.49 | 87.89 | 93.95 | 0.0778 |
| 3DPoV | DINOv2R-B/14 | **59.17** | **83.61** | **92.59** | **0.0933** | **69.61** | **88.47** | **94.19** | **0.0757** |

Table 3: Depth estimation results on Navi. Accuracy is reported using the threshold-based metrics $\delta_1$ ($< 1.25$), $\delta_2$ ($< 1.25^2$), and $\delta_3$ ($< 1.25^3$), as introduced by (Eigen et al., 2014). We also report RMSE in meters. Both scale-aware and scale-invariant scores are shown for completeness.

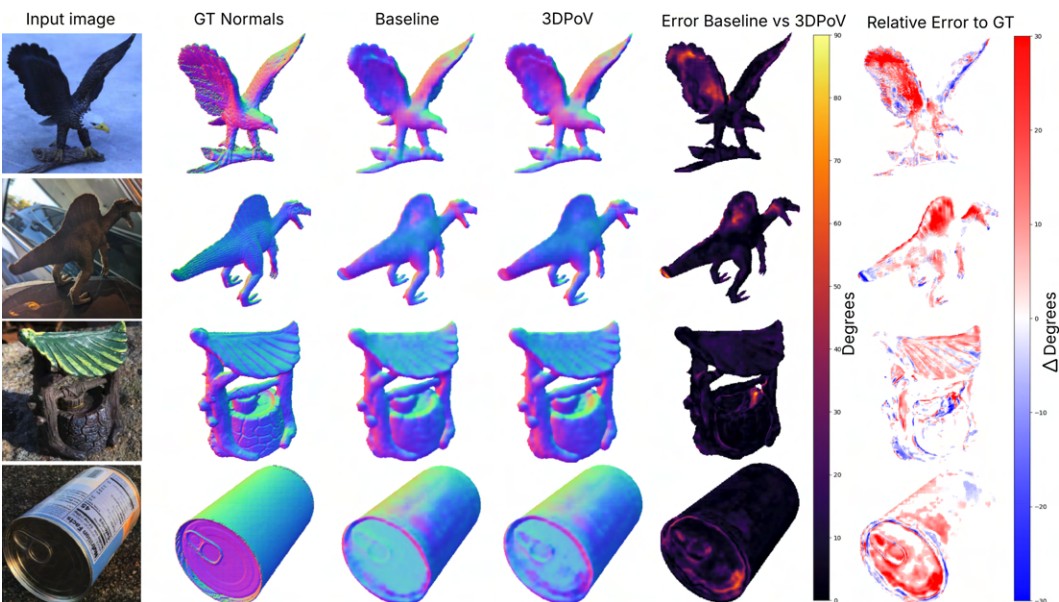

Figure 4: **Surface Normal Qualitative Examples.** To highlight differences between models with shared architecture, we visualize the angular error between the Baseline and 3DPoV predictions, which highlights regions where surface normal estimates differ. $\Delta$Error to GT denotes the difference in angular error between baseline and 3DPoV predictions with respect to the ground truth normals, shown only in regions of disagreement (error $> 5°$).Red areas indicate where 3DPoV predictions align more closely with the ground truth, while blue areas indicate where the baseline is closer. For baseline we use DinoV2 with Registers.

For surface normals (Figure 4), we visualize both predictions and angular error maps to highlight regions of divergence between 3DPoV and the baseline. 3DPoV provides more faithful orientation estimates, particularly under challenging conditions: on the eagle, it better recovers fine structure along the body and wing edges, and in the can example it reduces errors caused by reflective surfaces. These qualitative trends are consistent with the quantitative improvements reported in Table 4. Additional visualizations, including relative error maps with respect to ground truth, are provided in Figure 8.

## 5 ABLATION STUDIES

We conduct ablation studies to isolate the impact of key design choices in **3DPoV**. All experiments are based on the DinoV2-Reg backbone. To ensure fair comparisons, we vary only one factor per experiment and report performance at matched training durations.

**Reference extraction.** As shown in Table 5a, student-extracted references improve average

| Model | Backbone | 11.25° ↑ | 22.5° ↑ | 30° ↑ | RMSE ↓ |
|---|---|---|---|---|---|
| Dino | ViT-B/16 | 31.47 | 58.61 | 70.62 | 31.83 |
| 3DPoV | DINOv1-B/16 | **31.67** | **58.82** | **70.71** | **31.78** |
| DinoV2R | ViT-B14 | 37.10 | 65.93 | 77.09 | 28.07 |
| 3DPoV | DINOv2R-B/14 | **38.29** | **67.00** | **77.86** | **27.78** |

Table 4: Surface normal estimation results on Navi. We report accuracy at angular thresholds as well as the RMSE in degrees between predicted and GT normals.

performance by +2.37%, with the largest gains
on small viewpoint differences. This supports our design choice of letting the student process references, as it encourages more discriminative and sortable features.

**Number of frames.**    Table 5b compares training with 1, 2, and 4 frames. Since our flow does not intrinsically operate on a single frame, the 1-frame setup leverages NeCo-style overlapping crops as a proxy. Performance improves steadily with more frames: 4 frames bring +1.01% over 1 frame and +0.13% over 2 frames, indicating that temporal supervision benefits from richer context across all viewpoint regimes.

**Dataset choice.**    Table 5c shows that the CO3-YT-DL mixture outperforms any single dataset, confirming that diversity is key to robustness. While CO3D alone yields the weakest overall scores, adding it to YT–DL still improves the large-viewpoint bin (+0.26), highlighting that object-centric multiview footage provides complementary signal.

**Step size.**    Varying the temporal step between frames (Table 5d) shows that step 2 captures too little variation, while step 6 reduces visibility in multi-view datasets like CO3D and DL3DV, biasing tracks towards uninformative regions (sky/ground). Step 4 achieves the best trade-off, maintaining many visible points while capturing meaningful viewpoint changes.

**Resampling strategy.**    Latent-space interpolation (Table 5e) outperforms resized sampling (+0.36 overall), suggesting that operating directly in the feature grid avoids artifacts from upsampling and preserves finer spatial detail.

**Point tracker.**    Table 5f compares RAFT (Teed & Deng, 2020) and CoTrackerV3 (Karaev et al., 2024). Our method improves with both, showing independence from tracker choice, but CoTrackerV3 performs best (+0.33 overall), likely due to its robustness to occlusions and sudden motion compared to optical flow methods.

Table 5: **Ablation of Key Design Choices in 3DPoV.** We report Keypoint Matching Recall on SPair-71k across viewpoint difficulty levels–Small, Medium, Large, and All.

(a) References Extracted by

| MODEL | S / 0 | M / 1 | L /2 | ALL |
|---|---|---|---|---|
| Teacher | 57.95 | 51.04 | 53.05 | 53.03 |
| Student | **60.16** | **52.79** | **54.50** | **55.40** |

(b) Number of frames

| FRAMES | S / 0 | M / 1 | L /2 | ALL |
|---|---|---|---|---|
| 1 | 59.26 | 52.02 | 54.00 | 54.39 |
| 2 | 60.04 | 52.72 | 54.35 | 55.27 |
| 4 | **60.16** | **52.79** | **54.50** | **55.40** |

(c) Dataset choice

| DATA | S / 0 | M / 1 | L /2 | ALL |
|---|---|---|---|---|
| CO3D | 59.32 | 52.29 | 54.08 | 54.58 |
| YTVoS | 59.71 | 52.56 | 54.34 | 55.02 |
| DL3DV | 59.84 | 52.41 | 54.24 | 55.02 |
| YT-DL | 60.02 | 52.66 | 54.24 | 55.25 |
| CO3-YT-DL | **60.16** | **52.79** | **54.50** | **55.40** |

(d) Step size on frame sampling

| STEP | S / 0 | M / 1 | L /2 | ALL |
|---|---|---|---|---|
| 2 | 60.04 | 52.75 | 54.50 | 55.22 |
| 4 | **60.16** | **52.79** | **54.50** | **55.40** |
| 6 | 59.70 | 52.28 | 54.06 | 55.02 |

(e) Type of Resampling

| METHOD | S / 0 | M / 1 | L /2 | ALL |
|---|---|---|---|---|
| Resized | 59.83 | 52.42 | 54.20 | 55.04 |
| Latent | **60.16** | **52.79** | **54.50** | **55.40** |

(f) Choice of Point Tracker

| TRACKER | S / 0 | M / 1 | L /2 | ALL |
|---|---|---|---|---|
| RAFT | 59.81 | 52.43 | 54.31 | 55.07 |
| CoTrackerV3 | **60.16** | **52.79** | **54.50** | **55.40** |

## 6    CONCLUSION

In this paper, we introduced 3DPoV, a framework for learning dense, viewpoint-invariant features through temporally anchored permutation supervision. By integrating point tracks with reference-based sorting, our method enforces relative similarity structures that remain stable across time, occlusion, and viewpoint variation. Evaluations across Probe3D tasks demonstrate consistent improvements over all baselines, with balanced gains across both small and large viewpoint shifts and

emerging robustness to challenging lighting conditions. These results highlight the value of temporal ranking as a supervisory signal and suggest that point tracking can serve as a powerful tool for geometry-aware representation learning without requiring explicit 3D labels.

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

## A  DISCLOSURE OF LLM USAGE

We declare that the use of LLMs for writing this paper was limited to general-purpose writing assistance. Specifically, we used them only to polish the wording of text sections and in no way to generate the research ideas or technical results and proofs presented in this paper.

## B  RELAXED SORTING AND SOFT PERMUTATION MATRICES

Sorting is a non-differentiable operation, which prevents gradient-based optimization when comparing ranked outputs. Traditional sorting uses discrete element swaps, such as $d_i' \leftarrow \min(d_i, d_j)$, which introduce discontinuities. To enable smooth learning, we adopt a differentiable sorting approach that relaxes these comparisons into continuous, pairwise soft-sorting operations.

Following (Lee et al., 2017), for any pair of distances $d_i, d_j$ (drawn from a row of the distance matrix $D$), the relaxed sorting step is defined as:

$$\text{softmin}(d_i, d_j) = d_i f(d_j - d_i) + d_j f(d_i - d_j) \tag{8}$$
$$\text{softmax}(d_i, d_j) = d_i f(d_i - d_j) + d_j f(d_j - d_i) \tag{9}$$

where $f(x) = \frac{1}{\pi} \arctan(\beta x) + 0.5$ is a sigmoid-shaped function centered at $x = 0$, and $\beta > 0$ controls the steepness of the relaxation.

As $\beta \to \infty$, the function $f(x)$ approaches a step function, and the sorting converges to discrete behavior. In practice, we use moderate values ($\beta = 3$ or $20$), which result in **soft permutations** that retain uncertainty and allow smooth gradient flow–ideal for ambiguous or occluded regions in video.

These pairwise comparisons are composed into **elementary swap matrices** $P_{\text{swap}}(d_i, d_j) \in \mathbb{R}^{np_r \times np_r}$, each being a near-identity matrix except for a $2 \times 2$ block that softly mixes elements $i$ and $j$. The full differentiable sorting process applies a sequence of these swaps using the Odd-Even Sorting Network (Petersen et al., 2022):

$$P_t = \prod_{(i,j) \in \mathcal{M}_t} P_{\text{swap}}(d_i, d_j), \quad \mathcal{M}_t = \begin{cases} \text{odd indices}, & \text{if } t \text{ odd} \\ \text{even indices}, & \text{if } t \text{ even} \end{cases} \tag{10}$$

After $L = np_r$ steps, the final soft permutation matrix is obtained by composing all swap layers:

$$P = \prod_{t=1}^{L} P_t \in \mathbb{R}^{np_r \times np_r} \tag{11}$$

Each $P$ matrix describes a probabilistic ranking over reference patches. Each row of $P$ encodes a distribution over rank positions for one reference patch, while each column reflects the expected occupant of that rank. This soft structure captures a smooth approximation of the discrete sorting behavior.

In our implementation, we apply this procedure independently to each query patch. The resulting permutation matrices for a batch of size $B$ with $np_q$ query patches form a tensor:

$$P \in \mathbb{R}^{B \times np_q \times np_r \times np_r} \tag{12}$$

These permutation matrices capture the relative ordering of reference patches with respect to each query patch and serve as the foundation for our temporal consistency loss.

## C  FURTHER DETAILS OF LOSS FORMULATION

*Our loss formulation is intentionally designed to strengthen spatial discrimination in patch correspondences. Concretely, we supervise student permutation distributions using a reversed cross-entropy of the form:*

$$\mathcal{L}_{CE}^i = - \sum_{j=1}^{np_r} P_{t,r}^S[i,j] \cdot \log \left( P_{0,r}^T[i,j] + \epsilon \right), \tag{13}$$

*where the student distribution acts as the weighing measure.*

*If we express cross-entropy in terms of KL divergence and entropy,*

$$CE(P_A, P_B) = KL(P_A \parallel P_B) + H(P_A), \tag{14}$$

*the direction used in prior work such as NeCo, $CE(P_T, P_S)$, reduces (up to constants) to minimizing $KL(P_T \parallel P_S)$ because $H(P_T)$ is fixed when the teacher is frozen (receives EMA updates).*

*This corresponds to a mode-covering divergence: the student must distribute mass wherever the teacher assigns probability, encouraging broad, soft distributions that cover the teacher's uncertainty.*

*In contrast, the loss we apply, $CE(P_S, P_T)$, can be expressed as :*

$$CE(P_S, P_T) = KL(P_S \parallel P_T) + H(P_S), \tag{15}$$

*where $H(P_S)$ is not constant. Minimizing this loss therefore simultaneously reduces $KL(P_S \parallel P_T)$ while suppressing the entropy of the student, promoting high-confidence, sharply peaked ranking distributions.*

*The optimum of this loss is a deterministic distribution that assigns all mass to the teacher's highest-probability candidate, illustrating its mode-seeking nature. Thus, in ambiguous correspondence cases, our formulation encourages the student to make confident, spatially discriminative predictions rather than reproducing the teacher's diffuse uncertainty.*

Figure 5 illustrates how multiple reference frames contribute to the loss.

## D  DATASET CHOICE

The choice of fine-tuning data significantly shapes the model's capacity to learn meaningful correspondences and geometric understanding from videos. Using video as a modality introduces variability along several axes: camera motion (static vs. dynamic), object movement, scene composition, occlusion patterns, viewpoint shifts, and lighting conditions. Capturing this diversity is essential for enhancing dense self-supervised learning, particularly when supervision operates at the level of patch correspondences and temporal consistency.

To this end, we fine-tune on a blend of complementary datasets, each contributing to different facets of the video distribution. For learning object-centered 3D structure and viewpoint-invariant patterns, we rely on CO3D (Reizenstein et al., 2021), which provides long video sequences of individual objects viewed under large viewpoint variations, often spanning 180 degrees or more (Figure 9). The

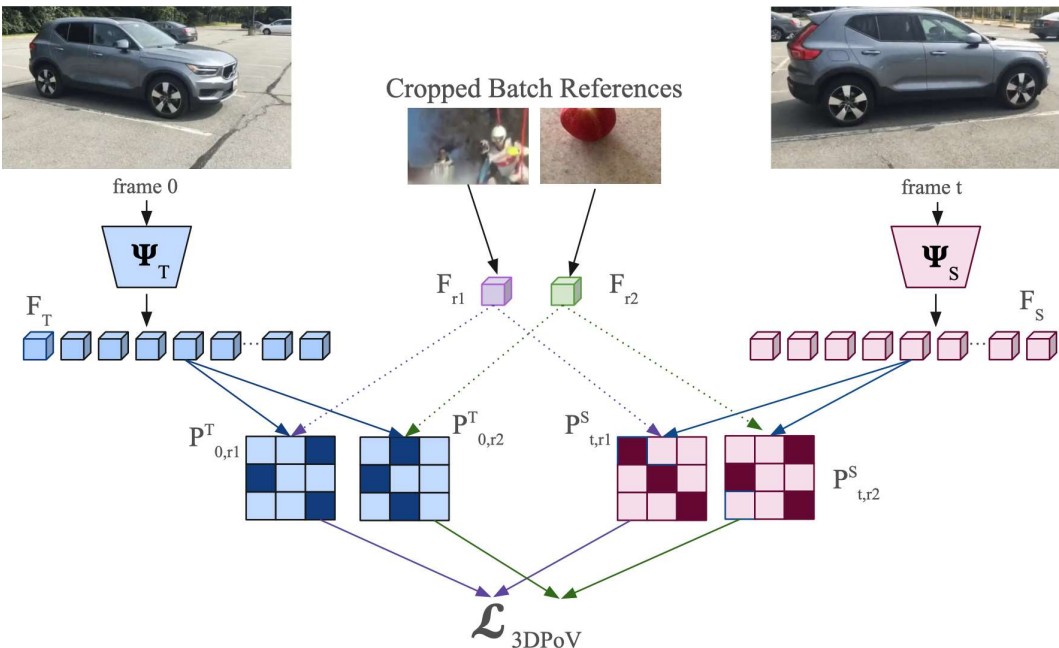

Figure 5: **Multiple reference contribution to the final loss.** Given two reference features $F_{r1}$ and $F_{r2}$ sampled from the feature bank, we compute corresponding permutation matrices $P_{0,r}^T$ and $P_{t,r}^S$ for each reference crop, comparing teacher (anchor frame $t = 0$) and student (future frame $t$) features. The permutation-based loss is computed for each reference independently by aligning the student and teacher permutations. The final loss is obtained by averaging over all such reference-specific losses.

dataset spans both indoor and outdoor contexts, and includes challenging factors such as occlusion, background clutter, and varying lighting conditions–making it especially well-suited for learning spatially consistent patch-level features under changing viewpoints and appearances.

For scene-level understanding, we incorporate DL3DV (Ling et al., 2024), a large-scale dataset of RGB-D video sequences captured using a commodity LiDAR-equipped phone. DL3DV contains over 10,000 dynamic scenes recorded in both indoor and outdoor settings, offering a wide range of spatial layouts and motion patterns (Figure 10). While we do not use depth annotations, the diversity in geometry and camera motion supports learning structure-aware features that generalize to complex 3D environments.

To encourage temporal coherence and robustness to real-world motion, we also train on YouTube-VOS (Xu et al., 2018), a large-scale video dataset containing high-resolution clips of everyday activities involving multiple objects, scene changes, occlusion events, and complex camera trajectories. These sequences provide valuable temporal signal, allowing the model to learn how to maintain patch-level consistency across time under natural, unconstrained motion.

Together, these datasets span a wide range of visual conditions–from single-object multiview videos to dynamic, cluttered scenes with complex motion. This diversity supports learning dense, geometry-aware representations that generalize across tasks such as surface normal estimation, depth prediction, and keypoint correspondence.

## E  EXPERIMENTAL SETUP

Following TimeTuning (Salehi et al., 2023), we initialize our models using publicly available pre-trained DINO backbones. Specifically, we experiment with ViT-Base backbones from DINOv1 (Caron et al., 2021), DINOv2 with Registers (Oquab et al., 2023) and DINOv3(Siméoni et al.,

2025). Unless otherwise stated, we use DINOv2R as our reference baseline and fine-tune only the final layer of the frozen backbone.

We train all models on 4 NVIDIA A100 GPUs using AdamW with cosine learning rate decay. For DINOv1-based variants, the feature extractor is updated with a learning rate of 1e-5 and the remaining layers with 1e-4, applying a weight decay of 1e-4. For DINOv2-reg models, which converge faster, we use 1e-7 for the extractor and 1e-6 for the rest of the model, with a weight decay of 1e-5. All DINOv3 experiments are conducted using the same training setup and evaluation protocol as DINOv2 to ensure comparability. DINOv3 experiments are performed under the same fine-tuning regime as DINOv2 to ensure comparability, though more tailored settings will be explored in future work.

Training is lightweight compared to large-scale pretraining: fine-tuning requires roughly 5 hours on 4×A100 GPUs ($\approx$20 GPU-hours) for 9,242 samples. For perspective, this is less than the cost of a single additional epoch of DINOv2 pretraining, which was conducted on 142M images and demanded multi-week training on large-scale GPU clusters. Thus, the reported improvements are achieved with a negligible fraction of the original pretraining cost. These configurations follow the same optimization strategy as MoSiC (Salehi et al., 2024), with adjustments tailored to each backbone variant.

### E.1  DATASET CONFIGURATION

Due to the substantial imbalance in dataset sizes, we subsampled CO3D to ensure a more even distribution of training samples across the three sources. Specifically, with a frame sampling step of 10, the full CO3D dataset yielded 16,345 samples, while YouTube-VOS and DL3DV provided only 3,471 and 1,150 samples, respectively. To avoid training bias, we reduced the CO3D sample count to match that of YouTube-VOS.

Additionally, to compensate for the lower volume and higher complexity of DL3DV scenes–often containing multiple objects and fine structural details–we applied two different preprocessing strategies. One variant followed the standard resizing pipeline used across all datasets (resizing to 224×224). The other employed a center crop to match the 224×224 resolution used in our training pipeline. This center crop was necessary to ensure frame alignment required by the tracking module, and it is particularly favorable for preserving spatial and scene-level details that could otherwise be degraded by uniform resizing. The final training distribution consisted of 3,471 samples from CO3D, 3,471 from YouTube-VOS, and 2,300 from DL3DV.

### E.2  KEYPOINT MATCHING

On **SPair-71k**, we follow the Probe3D protocol. Dense spatial features are extracted from both images in a pair, and cosine similarity is computed between all spatial locations. For each annotated keypoint in the source image, the target location is predicted as the position with the highest similarity. Recall is then computed based on the spatial distance between predicted and ground-truth keypoints at varying thresholds.

The benchmark categorizes pairs into three viewpoint groups (small, medium, large). The "All" split aggregates these categories and additionally includes pairs that do not fall into any viewpoint-defined subset. Due to the imbalance in dataset distribution, the "All" score is heavily influenced by small-viewpoint pairs and should not be interpreted as a direct average across difficulty regimes.

On **Navi**, evaluation leverages access to ground-truth 3D geometry and calibrated cameras. Following Probe3D, dense features are projected onto a 3D grid, and correspondences are established directly in 3D space. Performance is assessed in two complementary ways:

- **3D error** – the Euclidean distance between predicted and ground-truth 3D points, aligned into a shared coordinate frame using camera pose.

- **2D reprojection error** – the pixel-level distance between the reprojected 3D predictions and the ground-truth 2D keypoints.

We report recall at multiple thresholds (e.g., <2cm in 3D, <5px in 2D) and break down results by relative camera rotation. This dual evaluation provides a comprehensive test of whether features preserve geometric consistency across views.

### E.3 DEPTH ESTIMATION

Depth evaluation follows the protocol introduced by (Eigen et al., 2014), which includes both error-based and accuracy-based metrics. The primary error metric is the root mean squared error (RMSE), computed between the predicted depth values $d_{pred}$ and ground truth $d_{gt}$. In addition, accuracy is measured using threshold-based metrics defined as the percentage of pixels for which the ratio between prediction and ground truth is within a multiplicative threshold. More formally, accuracy at threshold is

$$\delta_i(d^{pr}, d^{gt}) = \frac{1}{N} \sum_{j \in N} max \left( \frac{d_j^{pr}}{d_j^{gt}}, \frac{d_j^{gt}}{d_j^{pr}} < 1.25^i \right) \tag{16}$$

where $i \in 1, 2, 3$. The thresholds $\delta_1, \delta_2, \delta_3$ therefore correspond to tolerance levels of $1.25$, $1.25^2$ and $1.25^3$ respectively.

For depth estimation, we report both scale-aware and scale-invariant metrics. The scale-aware RMSE (in meters) reflects absolute depth accuracy and is sensitive to global scale. In contrast, the scale-invariant RMSE normalizes per-frame predictions to account for scale ambiguity, capturing relative depth structure. Both are included for completeness.

As NAVI was not originally created as a depth benchmark, the authors of Probe3D adapt it by leveraging the underlying 3D geometry from multiview reconstructions to define a relative depth signal between pixels across view pairs. In this context, scale-invariant results are more aligned with the intent of the benchmark, as they emphasize relative spatial structure rather than absolute scale.

### E.4 SURFACE NORMAL ESTIMATION

For surface normal evaluation, we follow the setup described in .(Bae et al., 2021), where the goal is to assess the angular consistency between predicted normals $n_{pred}$ and ground truth normals $n_{gt}$. Specifically, we compute the angle $\theta$ between the two vectors at each pixel and report the percentage of pixels for which this angular error is below predefined thresholds. Following the benchmark, we report accuracy at $11.25°$, $22.5°$, $30°$ along with RMSE for the angular error.

## F FURTHER RESULTS

## G FURTHER ABLATIONS

For completeness, we also report ablation results on Navi keypoint matching in Table 7, complementing the SPair analysis presented in the main paper. The overall trends are consistent across the two benchmarks, confirming that our design choices generalize beyond 2D correspondence. On Navi, improvements under large viewpoint changes are smaller in magnitude compared to our preferred setup, yet the performance remains competitive. Taken together, the results across SPair and Navi highlight that 3DPoV delivers consistent benefits across both 2D and 3D correspondence evaluations.

*We introduce patches from batch clips (external reference) to ensure diversity in similarity values and scenes. This setup follows the configuration from NeCo. In contrast, the addition of crops from the same clip (internal reference) ensures high-similarity anchors within the broader distribution, sharpening the ranking and ensuring a positive signal for the gradient. Nonetheless, the use of internal crops is limited to the number of frames used in training. We ablated this design choice in Table 8, reducing the number of references to match the number of frames and observe that indeed exclusive use of internal references result in better performance (+0.32% on 'All'). This suggests*

| (a) 3D keypoint matching | | | | |
|---|---|---|---|---|
| Model | Backbone | 0.01m | 0.02m | 0.05m |
| Dino | ViT-S16 | **26.12** | **43.10** | **74.80** |
| TimeTuning | ViT-S16 | 24.17 | 41.44 | 73.36 |
| MoSiC | ViT-S16 | 23.69 | 40.94 | 72.98 |
| Dino | ViT-B16 | 26.12 | 43.10 | 74.80 |
| NeCo | ViT-B16 | 24.24 | 41.20 | 73.20 |
| 3DPoV | ViT-B16 | **26.52** | **43.53** | **74.99** |
| DinoV2-**reg** | ViT-B14 | 34.10 | 53.79 | 82.43 |
| MoSiC-**reg** | ViT-B14 | 33.26 | 53.24 | 82.53 |
| NeCo-**reg** | ViT-B14 | 31.70 | 51.13 | 81.22 |
| 3DPoV-**reg** | ViT-B14 | **34.82** | **54.39** | **82.56** |
| DinoV3-**reg** | ViT-B16 | 38.33 | **56.95** | 83.69 |
| 3DPoV-**reg** | ViT-B16 | **38.36** | 56.93 | **83.72** |

| (b) 2D keypoint matching | | | | |
|---|---|---|---|---|
| Model | Backbone | 5px | 25px | 50px |
| Dino | ViT-S16 | **3.47** | **22.69** | **37.49** |
| TimeTuning | ViT-S16 | 2.86 | 20.33 | 35.32 |
| MoSiC | ViT-S16 | 2.78 | 20.04 | 34.82 |
| Dino | ViT-B16 | 3.47 | 22.69 | 37.49 |
| NeCo | ViT-B16 | 3.18 | 21.05 | 35.68 |
| 3DPoV | ViT-B16 | **3.58** | **23.07** | **37.71** |
| DinoV2-**reg** | ViT-B14 | 4.34 | 30.28 | 48.00 |
| MoSiC-**reg** | ViT-B14 | 4.14 | 29.51 | 47.59 |
| NeCo-**reg** | ViT-B14 | 4.36 | 29.37 | 46.51 |
| 3DPoV-**reg** | ViT-B14 | **4.54** | **31.08** | **48.65** |
| DinoV3-**reg** | ViT-B16 | 5.76 | **36.68** | 53.44 |
| 3DPoV-**reg** | ViT-B16 | **5.77** | **36.68** | **53.46** |

Table 6: Comparison of Navi Recall for 3D (a) and 2D (b) keypoint matching at different thresholds. Higher is better.

Table 7: **Ablation of Key Design Choices in 3DPoV.** We report Keypoint Matching Recall on NAVI. Each experiment isolates one design parameter, with other settings held fixed.

| (a) References Extracted by | | | | |
|---|---|---|---|---|
| MODEL | $\theta_0^{15}$ | $\theta_{15}^{30}$ | $\theta_{30}^{60}$ | $\theta_{60}^{180}$ |
| Teacher | 87.56 | 67.61 | 47.10 | **31.35** |
| Student | **89.22** | **69.23** | **47.48** | 31.33 |

| (b) Number of frames - Use all frames | | | | |
|---|---|---|---|---|
| FRAMES | $\theta_0^{15}$ | $\theta_{15}^{30}$ | $\theta_{30}^{60}$ | $\theta_{60}^{180}$ |
| 1 | 88.47 | 68.34 | 47.46 | **31.52** |
| 2 | 89.12 | 69.17 | **47.51** | 31.42 |
| 4 | **89.22** | **69.23** | 47.48 | 31.33 |

| (c) Dataset choice - Navi eval | | | | |
|---|---|---|---|---|
| DATA | $\theta_0^{15}$ | $\theta_{15}^{30}$ | $\theta_{30}^{60}$ | $\theta_{60}^{180}$ |
| CO3D | 88.79 | 68.57 | 47.31 | 31.37 |
| YTVoS | 88.76 | 68.64 | 47.34 | **31.37** |
| DL3DV | 88.82 | 68.88 | **47.55** | 31.37 |
| YT-DL | 88.91 | 69.00 | **47.55** | 31.33 |
| CO3-YT-DL | **89.22** | **69.23** | 47.48 | 31.33 |

| (d) Step size on frame sampling | | | | |
|---|---|---|---|---|
| STEP SIZE | $\theta_0^{15}$ | $\theta_{15}^{30}$ | $\theta_{30}^{60}$ | $\theta_{60}^{180}$ |
| 2 | 89.13 | 69.20 | **47.54** | **31.42** |
| 4 | **89.22** | **69.23** | 47.48 | 31.33 |
| 6 | 88.87 | 68.81 | 47.32 | 31.24 |

| (e) Type of Resampling | | | | |
|---|---|---|---|---|
| RESAMPLING | $\theta_0^{15}$ | $\theta_{15}^{30}$ | $\theta_{30}^{60}$ | $\theta_{60}^{180}$ |
| Resized | 88.82 | 68.91 | 47.60 | **31.52** |
| Latent | **89.22** | **69.23** | **47.48** | 31.33 |

| (f) Choice of Point Tracker | | | | |
|---|---|---|---|---|
| TRACKER | $\theta_0^{15}$ | $\theta_{15}^{30}$ | $\theta_{30}^{60}$ | $\theta_{60}^{180}$ |
| RAFT | 88.91 | 69.03 | 47.46 | 31.37 |
| CoTrackerV3 | **89.22** | **69.23** | **47.48** | **31.33** |

*that internal reference patches maximize the coverage of the specific dynamic scene, which is more valuable for learning fine-grained 3D correspondence.*

| REFERENCES | S / 0 | M / 1 | L /2 | ALL |
|---|---|---|---|---|
| External (4) | 59.66 | 52.27 | 54.40 | 54.89 |
| Internal (1) + external (3) | 59.87 | 52.44 | 54.27 | 55.07 |
| Internal (4) | **60.24** | **52.71** | **54.36** | **55.39** |

Table 8: *SPair-71k keypoint matching ablations. For a reference pool of size 4 we compare different internal/external splits*

| TRACKER | S / 0 | M / 1 | L /2 | ALL |
|---|---|---|---|---|
| RAFT | 59.81 | 52.43 | 54.31 | 55.07 |
| CoTrackerV2 | 59.03 | 51.74 | 53.82 | 54.18 |
| CoTrackerV3 | **60.16** | **52.79** | **54.50** | **55.40** |

Table 9: *SPair-71k Keypoint Matching; Ablating choice of tracker*

*A further experiment focuses on removing the sorting algorithm, and simply applying the cross-entropy loss on the similarity matrices. The results are presented in Table 10*

| METHOD | S / 0 | M / 1 | L /2 | ALL |
|---|---|---|---|---|
| Similarity matrix | 59.49 | 52.36 | 54.42 | 54.81 |
| Sorted similarity matrix | **60.16** | **52.79** | **54.50** | **55.40** |

Table 10: *SPair-71k Keypoint Matching; Ablating sorting module by removing the differentiable sorting module and leverage the similarity matrices directly*

| UNFROZEN BLOCKS | S / 0 | M / 1 | L /2 | ALL |
|---|---|---|---|---|
| Blocks 8-11 | 57.66 | 49.84 | 51.48 | 52.50 |
| Blocks 10-11 | **60.16** | **52.79** | **54.50** | **55.40** |
| Block 11 | 58.64 | 51.72 | 53.41 | 53.79 |

Table 11: *SPair-71k Keypoint Matching; we unfreeze a number of layers and experiment under the same setup*

*Our method assigns the Teacher to the first frame ($t = 0$) and the Student to subsequent frames. We tested the reverse configuration Table 12. The results confirm our design choice (55.40% vs 55.05%), where CoTracker initializes points at $t = 0$, guaranteeing they are visible and unoccluded. Assigning the Teacher to $t = 0$ ensures the target features are reliable. Using later frames as the anchor introduces occlusion noise into the supervision signal.*

| TEACHER FRAME | S / 0 | M / 1 | L /2 | ALL |
|---|---|---|---|---|
| Last (nf-1) | 59.85 | 52.40 | 54.28 | 55.05 |
| First (0) | **60.16** | **52.79** | **54.50** | **55.40** |

Table 12: *SPair-71k Keypoint Matching; we ablate the choice of frame being processed by the teacher and consequently used as anchor.*

| NO. OF REF | S / 0 | M / 1 | L /2 | ALL |
|---|---|---|---|---|
| 3 | 59.94 | 52.43 | 54.20 | 55.10 |
| 4 | 59.87 | 52.44 | 54.27 | 55.07 |
| 5 | **60.16** | **52.79** | **54.50** | **55.40** |
| 7 | 59.84 | 52.37 | 54.28 | 55.04 |

Table 13: *SPair-71k keypoint matching ablations. Ablating number of references*

## H MAPPING TO PROBE3D BENCHMARK

We compare our reproduced baselines and reported results with the original Probe3D study in Table 15, Table 16, Table 17, Table 18. Minor misalignments are expected due to differences in environment and training setup, but overall trends are consistent with the original benchmark.

| Loss | S / 0 | M / 1 | L /2 | All |
|---|---|---|---|---|
| $CE(t,s)$ | 58.17 | 51.29 | 53.13 | 53.27 |
| $CE(s,t)$ | **60.16** | **52.79** | **54.50** | **55.40** |

Table 14: *SPair-71k Keypoint Matching; ablation on loss direction*

| Model | Backbone | Data | S / 0 | M / 1 | L /2 | All |
|---|---|---|---|---|---|---|
| Dino † | ViT-B16 | IN-1k | 30.4 | 24.0 | 24.3 | 26.8 |
| Dino | ViT-B16 | IN-1k | 30.19 | 24.22 | 24.35 | 26.39 |
| 3DPoV | ViT-B16 | CO3-YT-DL | **31.66** | **25.74** | **25.94** | **28.12** |
| DinoV2-**reg**† | ViT-B14 | LVD | 58.3 | 51.4 | 53.4 | 53.7 |
| DinoV2-**reg** | ViT-B14 | LVD | 58.20 | 51.56 | 53.41 | 53.47 |
| 3DPoV | ViT-B14 | CO3-YT-DL | **60.16** | **52.79** | **54.50** | **55.40** |

Table 15: SPair-71k viewpoint difference. 0: No significant view difference (same view or minimal changes), 1: Moderate viewpoint difference, 2: Large viewpoint difference. Here † represents the bechmark reported values

| Model | Backbone | Data | $\theta_0^{15}$ | $\theta_{15}^{30}$ | $\theta_{30}^{60}$ | $\theta_{60}^{180}$ |
|---|---|---|---|---|---|---|
| Dino† | ViT-B16 | IN-1k | 86.0 | 56.0 | 31.3 | 20.3 |
| Dino | ViT-B16 | IN-1k | 86.13 | 56.92 | 33.37 | 19.74 |
| 3DPoV | ViT-B16 | CO3-YT-DL | **86.42** | **57.18** | **33.77** | **20.42** |
| DinoV2-**reg**† | ViT-B14 | LVD | 89.0 | 67.3 | 44.8 | 31.1 |
| DinoV2-**reg** | ViT-B14 | LVD | 87.92 | 67.74 | 47.18 | **31.57** |
| 3DPoV | ViT-B14 | CO3-YT-DL | **89.22** | **69.23** | **47.48** | 31.33 |

Table 16: Navi Performance. Here † represents the bechmark reported values.

| Model | Backbone | Scale-Aware | | | | Scale-Invariant | | | |
|---|---|---|---|---|---|---|---|---|---|
| | | $\delta_1\uparrow$ | $\delta_2\uparrow$ | $\delta_3\uparrow$ | RMSE $\downarrow$ | $\delta_1\uparrow$ | $\delta_2\uparrow$ | $\delta_3\uparrow$ | RMSE $\downarrow$ |
| DinoV2-**reg**† | ViT-B/14 | - | - | - | - | 66.56 | 87.94 | 94.74 | 0.0806 |
| DinoV2-**reg** | ViT-B14 | 57.62 | 82.49 | 91.97 | 0.0960 | 68.49 | 87.89 | 93.95 | 0.0778 |
| 3DPoV-**reg** | ViT-B14 | **59.17** | **83.61** | **92.59** | **0.0933** | **69.61** | **88.47** | **94.19** | **0.0757** |

Table 17: Depth estimation results on Navi.

| Model | Backbone | 11.25° ↑ | 22.5° ↑ | 30° ↑ | RMSE ↓ |
|---|---|---|---|---|---|
| DinoV2-**reg**† | ViT-B/14 | 45.81 | 72.00 | 81.28 | 25.66 |
| DinoV2--**reg** | ViT-B14 | 37.10 | 65.93 | 77.09 | 28.0693 |
| 3DPoV-**reg** | ViT-B14 | **38.29** | **67.00** | **77.86** | **27.7798** |

Table 18: Surface normal estimation results on Navi.

# I COMPARING WITH THE MOST SIMILAR MODEL

During our ablation studies, we adopted an image processing strategy similar to NeCo–cropping frames followed by ROI alignment of the crops. This defines the 3DPoV-1frame experiment. As shown in Table 19, when compared directly to the baseline and NeCo, our approach demonstrates stronger ability to learn robust 3D representations, particularly under medium and large viewpoint shifts. This trend is consistent with the central challenge emphasized by the Probe3D benchmark, where performance typically drops sharply at larger viewpoint changes. We also note that the 'All' score–an aggregate over all categories, including samples not belonging to any category–is biased toward easier (small-shift) cases, and therefore differs in interpretation from a category-wise average.

| Model | Backbone | Data | S / 0 | M / 1 | L /2 | All |
|---|---|---|---|---|---|---|
| DinoV2-**reg** | ViT-B14 | LVD | 58.20 | 51.56 | 53.41 | 53.47 |
| NeCo-**reg** | ViT-B14 | COCO | **59.57** | 49.06 | 52.35 | 54.42 |
| 3DPoV-1Frame-**reg** | ViT-B14 | CO3-YT-DL | 59.49 | **52.18** | **54.22** | **54.66** |

Table 19: SPair71k Keypoint Matching results. Compared to the most relevant prior method (NeCo), 3DPoV attains similar performance in the 'All' category while offering improvements in the more challenging Medium and Large viewpoint shift categories.

## J    FAILURE CASES

*Since our method inherits correspondences from CoTracker, its limitations can influence supervision quality. We therefore explore such cases in this section.*

*A representative scenario is shown in Figure 6, where multiple similar subjects (e.g., several blue fish in blue water) move in and out of frame. When the tracked fish exits the view, some points "jump" to a visually similar fish, and the tracker is unable to recover the original correspondence once it reappears. This produces an incorrect trajectory rather than a complete tracking loss.*

*3DPoV does not fully correct such failures, but its visibility-aware weighting reduces their impact. When a point drifts or becomes unreliable, its predicted visibility decreases, naturally lowering its contribution in the loss. As a result, these ambiguous temporal matches influence supervision less strongly, instead of being propagated as confident signals.*

*While this mechanism improves robustness in cluttered or out-of-frame motion, scenes with persistent ambiguity across many frames (e.g., long occlusions or repeated textures) remain challenging and are an interesting direction for future improvement.*

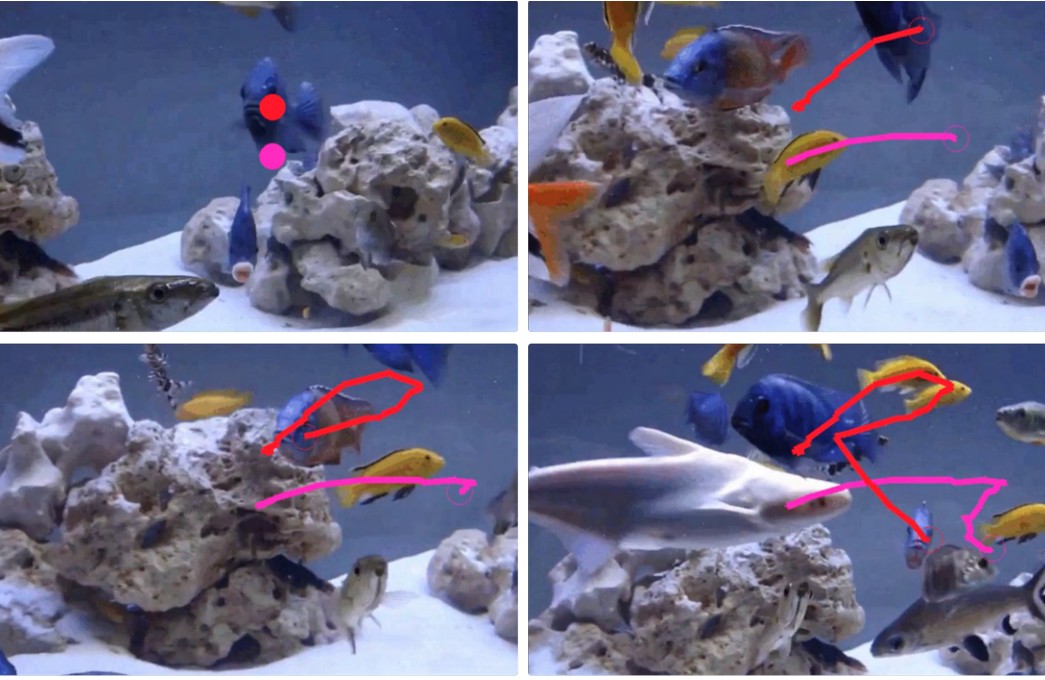

Figure 6: *Example of failure cases for tracking. Here we observe how the red tracked point jumps on similar pixels once the subject gets out of frame. The tracked dot now shows only the contour, indicating reduced visibility value which results in less weight during matching*

## K    ADDITIONAL VISUALIZATIONS

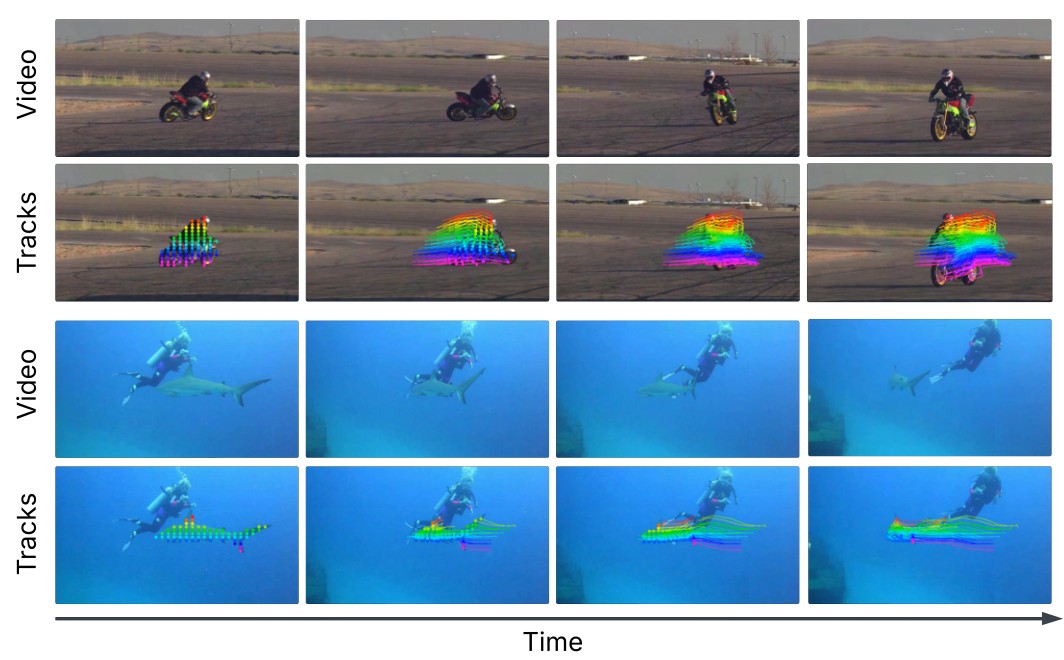

Figure 7: Examples of YTVoS movements and tracking quality

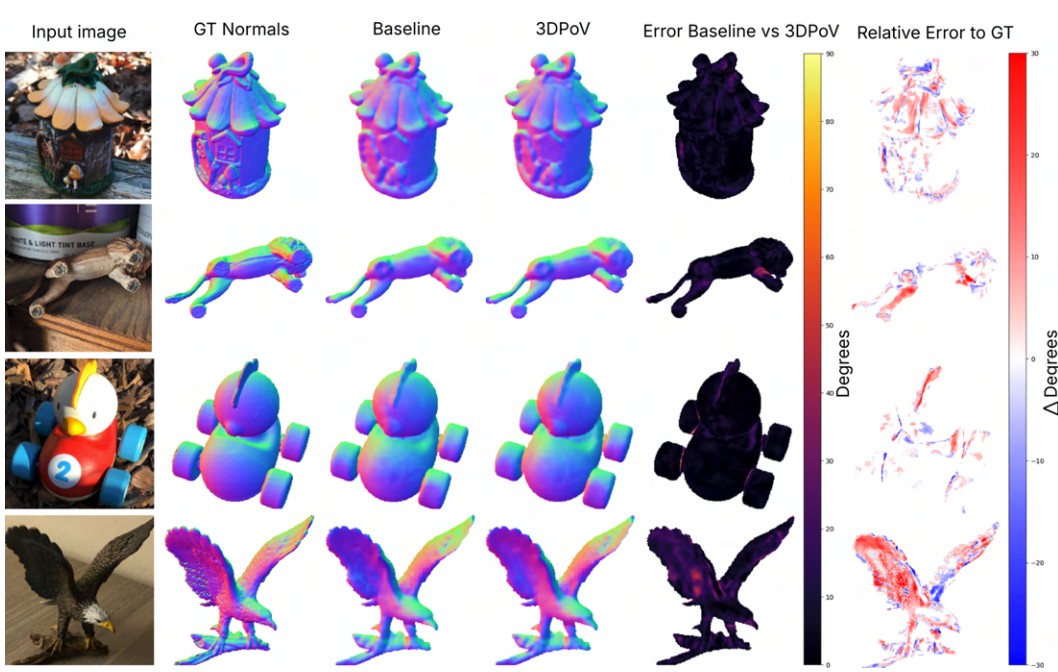

Figure 8: **More examples of surface normal qualitative results**

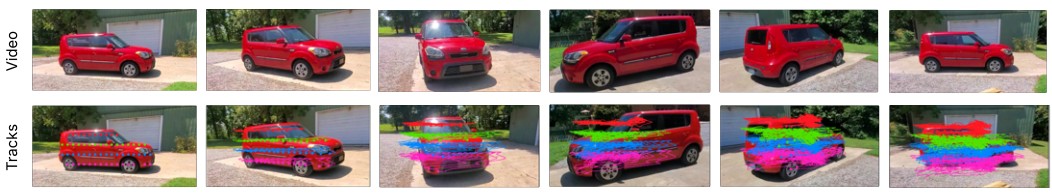

Figure 9: **More samples of CO3D movements and tracks**

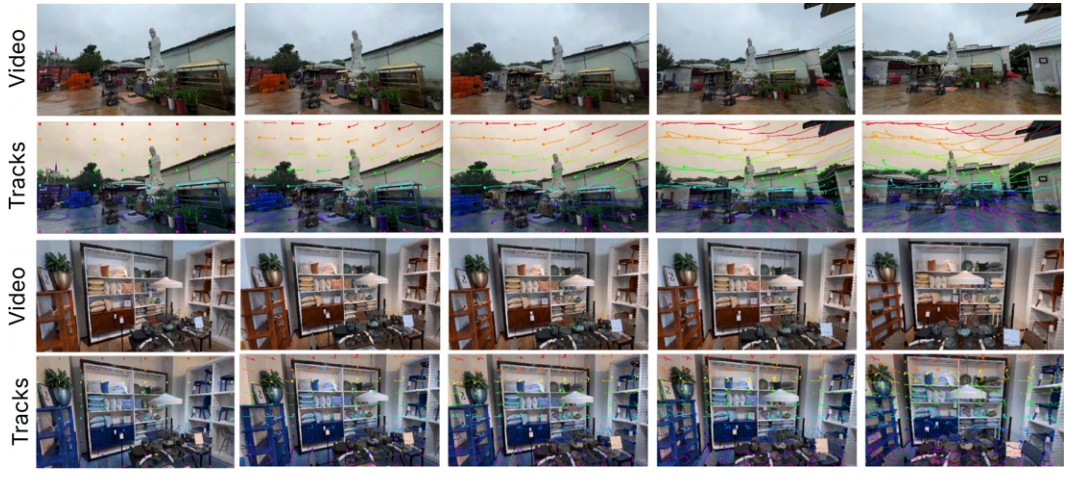

Figure 10: **Tracking behaviour acros DL3DV samples**

