# OpenReview forum: "3DPoV: Improving 3D understanding via Patch Ordering on Videos"
_ICLR.cc/2026/Conference — Submitted to ICLR 2026_

### Official Review · Reviewer_KGxc · 2025-10-26

**Soundness:** 2
**Presentation:** 3
**Contribution:** 3
**Rating:** 4
**Confidence:** 4

**Summary:**

The work develops a training framework to enhance the feature representations of self-supervised backbones (DINO, DINOv2) on 3D awareness tasks, such as multi-view correspondences, surface normal estimation and depth estimation. The framework trains on videos and uses point tracking as an additional cue for constructing the training objective. Relying on the student-teacher framework, the approach creates the training signal by computing a soft permutation matrix from the teacher and minimising the difference w.r.t. the corresponding matrix derived from the student network. The permutation matrix encodes the ordering of feature similarities: a query point, tracked in time, is compared to the “reference” points extracted from other videos and frames. The query points in the teacher and the student network may come from different frames in the video, but related to the same physical point, as predicted by a point tracker.

The approach trains only the final layer of the backbone network. The empirical results show benefit of the proposed training methodology across three tasks (depth, surface normals and multi-view consistency).

**Strengths:**

This is an interesting work with a few strong points to highlight:
* The work explores an exciting combination of point tracking and representation learning for 3D awareness.
* The proposed training strategy is new and interesting, yielding consistent benefits across the evaluated tasks. The results are particularly strong on keypoint matching.
* The evaluation included two backbone networks and explors a combination of three pre-training datasets.

**Weaknesses:**

* The empirical results are a bit mixed. While the approach does show a significant improvement on the SPair-71k dataset for multi-view correspondence, the benefit on depth and normal estimation is very marginal. The ablation study reveals further that point tracking yields marginal improvements compared to the use of optical flow. The experiment “Number of frames” (c.f. Tab. 5b) also undermines the need for point-tracking: considering only two frames already leads to a boost in downstream accuracy, on par with the setting using a larger temporal context. Overall, the experiments do not yet compellingly show a benefit of a longer temporal context and point tracking.
* The framework design is non-trivial, but the ablation study does not elaborate on key design choices. For example, differentiable sorting is central to the training objective, but there is no empirical study to show it is more effective than simpler alternatives (e.g. contrastive loss, Sinkhorn-Knopp). Another example is the choice of the student and teacher features: one is taken from the future frames, while the other is from t = 0. A third example are the reference crops: how sensitive is the approach to the number of the reference crops?
* The evaluation setting is somewhat unclear. Probe3D trains a DPT decoder for surface normal and depth estimation. The DPT decoder has skip connections to indermediate layers of the encoder. If the DPT is used here, this implies that only a single skip connection is affected, since the work trains only the final layer of the base model. This creates a strong constraint on the potential improvement, and is unlikely to affect the downstream accuracy in a significant way.

**Questions:**

* What is the difference between 3DPoV/ViT-B16 in the tables (e.g. 6th and 12th row in Tab. 1)? Is it a typo and one of them should be ViT-S16?
* How many parameters are finetuned in the base network? What happens if we increase the number of the finetuned layers?
* Please clarify the evaluation protocol (see the note on the DPT decoder in “Weaknesses”).

---

> ### Author Response · Authors · 2025-11-22
>
> ## **1. Importance of videos/point trackers**
> - ### **1.1.The significance of results for depth estimation and surface normal  estimation.**
>
> We appreciate the reviewer’s concern and clarify the significance of our improvements from **Table 3 and 4** in the main paper. As shown, 3DPoV improves DINOv2-reg on scale-aware depth estimation by 0.6%–1.5% across different $\delta$, and by 0.2%–1.1% on scale-invariant evaluations. It also consistently reduces RMSE in both settings. Importantly, these gains are achieved while our finetuning amounts to only 0.34% of a single DINOv2-reg epoch (13 epochs of 9,242 samples using 4 frames compared to 142M).
>
> ---
> - ### **1.2. Clarifying the impact of using more frames and employing point-trackers.**
>
> We ablate the effect of increasing the number of input frames--from a single frame (purely image-based) up to four frames (video-based)--in Table 5(b) of the paper. For convenience, we repeat the results here. The trend is clear: adding more frames consistently improves 3D understanding. The performance gain from 1$\rightarrow$2 frames is substantial(0.88% for All), while the improvement from 2$\rightarrow$4 frames is smaller. The ablation on point-trackers shows that our method is robust across different tracker choices: even a simple tracker like RAFT yields clear improvements, while the strongest results are obtained with the more accurate CoTrackerV3. The reason for RAFT’s high performance is the scarcity of examples in our dataset where the object disappears or becomes occluded, cases in which RAFT typically fails.
>
> _Table 1: SPair-71k Keypoint Matching;  Ablating number of frames_
> |**No. of frames**|**S / 0**|**M / 1**|**L / 2** |**All**|
> |---|---|---|---|---|
> |1| 59.26| 52.02| 54.00|54.39|
> |2| 60.04| 52.72| 54.35|55.27|
> |**4**|**60.16**|**52.79**|**54.50**|**55.40**|
>
> ---
> ## **2. Justification of Key Design Choices**
> - ### **2.1.Differentiable sorting mechanism**
>
> Thank you for highlighting this interesting potential ablation. We have now conducted an additional experiment where we remove the sorting algorithm, and simply apply the cross-entropy loss on the similarity matrices. The results are presented below:
>
> _Table 2: SPair-71k Keypoint Matching; Ablating sorting module by removing the differentiable sorting and leverage the similarity matrices directly_
> |**Method**|**S / 0**|**M / 1**|**L / 2**|**All**|
> |---|---|---|---|---|
> |Similarity matrix|59.49|52.36|54.42|54.81|
> |**Sorted similarity matrix**|**60.16**|**52.79**|**54.50**|**55.40**|
>
> Standard contrastive/similarity losses tend to treat supervision as binary (match vs. non-match). In contrast, differentiable sorting provides graded supervision: it enforces a specific relative ranking among multiple candidates. This preserves fine-grained geometric distinctions that are lost when simply maximizing raw similarity.
>
> ---
> - ### **2.2.Choice of student and teacher features**
>
> Our method assigns the Teacher to the first frame ($t=0$) and the Student to subsequent frames. We tested the reverse configuration (Table below). The results confirm our design choice (55.40% vs 55.05%), where CoTracker initialises points at $t=0$, guaranteeing they are visible and unoccluded. Assigning the Teacher to $t=0$ ensures the target features are reliable. Using later frames as the anchor introduces occlusion noise into the supervision signal.
>
> _Table 3: SPair-71k Keypoint Matching; we ablate the choice of frame being processed by the teacher and consequently used as anchor._
> |**Teacher frame**|**S / 0**|**M / 1**|**L / 2**|**All**|
> |---|---|---|---|---|
> |Last (nf-1)|59.85|52.40|54.28|55.05|
> |**First (0)**|**60.16**|**52.79**|**54.50**|**55.40**|

---

> ### Author Response · Authors · 2025-11-22
>
> ## **2. Justification of Key Design Choices (cont.)**
> - ### **2.3.Number of reference frames**
>
> We analyse 3dPoV's  sensitivity to the size and source of the reference pool. From the table below, we observe that the performance is relatively stable with respect to pool size, peaking at 5 reference frames. Adding more frames (up to 7) yields diminishing returns, suggesting 5 frames captures the necessary distribution without adding noise.
>
> _Table 4: SPair-71k Keypoint Matching; Ablating number of references_
> |**No. of ref**|**S / 0**|**M / 1**|**L / 2**|**All**|
> |---|---|---|---|---|
> |3|59.94|52.43|54.20|55.10|
> |4|59.87|52.44|54.27|55.07|
> |**5**|**60.16**|**52.79**|**54.50**|**55.40**|
> |7|59.84|52.37|54.28|55.04|
>
> Additionally, we also find that the source of references is more critical than the number. In the Table below, we find that using "Internal" references (patches from the same video) consistently performs better than using only "External" ones. This indicates that "hard negatives" from the same dynamic scene are more valuable for learning 3D consistency than random patches from other scenes.
>
> _Table 5: SPair-71k Keypoint Matching; For a reference pool of size 4 we compare different internal/external splits_
> |**References**|**S / 0**|**M / 1**|**L / 2**|**All**|
> |---|---|---|---|---|
> |External (4)|59.66|52.27|54.40|54.89|
> |Internal (1) + external (3)|59.87|52.44|54.27|55.07|
> |**Internal (4)**|**60.24**|**52.71**|**54.36**|**55.39**|
>
> ---
> ### **3. On Surface Normal/Depth Estimation improvements and probe3D**
>
> We thank the reviewer for catching this detail and apologize for the confusion. The paper incorrectly states that we fine-tune only the last layer. In practice, we unfreeze the last two transformer layers (layer 10 and 11), and we will correct this in the final version. Ablation is shown in table Table 6 and we kindly ask the reviewer to refer to the full discussion provided in our response to **R-sdbB**.
>
> The DPT decoder aggregates features from multiple stages (low-level spatial details + high-level semantic context). By tuning Blocks 10–11, we explicitly refine the highest-level feature map fed into the DPT. The frozen lower layers provide robust spatial priors, which DINO captures effectively out-of-the-box. The errors 3DPoV targets such as temporal flickering or inconsistencies in reflective/ambiguous regions -- are high-level semantic failures, not low-level texture failures. By refining the last two layers, we provide the DPT with a temporally consistent global context. The decoder then successfully fuses this improved geometric context with the sharp spatial details from the frozen lower layers to produce better depth and normal maps.
>
> Our results show that attempting to unfreeze lower layers actually degrades performance (see table below), as it disrupts the spatial priors the DPT relies on.
>
> _Table 6: SPair-71k Keypoint Matching; we unfreeze a number of layers and experiment under the same setup_
> |**Unfrozen Blocks**|**S / 0**|**M / 1**|**L / 2**|**All**|
> |----|---|---|---|---|
> |Blocks 8-11 | 57.66| 49.84| 51.48| 52.50|
> |**Blocks 10-11**|**60.16**|**52.79**|**54.50**|**55.40**|
> |Blocks 11|58.64|51.72|53.41|53.79|

---

> ### Author Response · Authors · 2025-11-28
>
> Dear Reviewer KGxc,
>
> Thank you once again for taking the time and effort to review our submission. As the deadline for the discussion phase is approaching, we wanted to kindly check if you have any additional questions or concerns that we could address in our rebuttal.
>
> We appreciate your insights and look forward to your feedback!
>
> Best regards,
>
> The Authors

---

### Official Review · Reviewer_gkoF · 2025-11-01

**Soundness:** 3
**Presentation:** 4
**Contribution:** 3
**Rating:** 4
**Confidence:** 4

**Summary:**

The paper proposes 3DPoV, a self supervised post training objective that improves 3D awareness in vision backbones by aligning the ranking of patch similarities across video frames using differentiable sorting. A teacher student design with tracked points supplies temporally consistent and viewpoint tolerant signals without 3D labels. The method reports consistent gains on Probe3D keypoint matching under viewpoint changes and on linear probes for depth and normals.

**Strengths:**

- Originality: Ranking based temporal supervision on patch similarities is a fresh alternative to direct feature matching and leverages point tracks effectively.

- Quality: Results are consistent across multiple backbones with focused ablations on frames, datasets, and tracker choice.

- Clarity: The pipeline and loss are clearly specified with equations and a reasonable visibility treatment.

- Significance: Consistent gains on viewpoint sensitive benchmarks indicate that 3DPoV injects geometry aware structure into widely used backbones, which is practically valuable.

**Weaknesses:**

- **Sensitivity to tracking and visibility**: The approach depends on point tracks and learned visibility; robustness under drift, rapid motion, or occlusion could be analyzed more systematically.
- **Scope beyond Probe3D**: The results emphasize Probe3D; adding small but representative downstream tasks such as camera pose estimation or robotic correspondence would strengthen external validity.
- **Missing baselines and related works**: How does the method compared with "Multiview Equivariance Improves 3D Correspondence Understanding with Minimal Feature Finetuning" and "Omniview-tuning: Boosting viewpoint invariance of vision-language pre-training models", which also tried to improve the 3D understanding of DINO.
- **Results are a bit marginal**: Though there are consistent improvements over different datasets, they are a bit marginal and the method does not seem to gain much with a heavy video guided correspondence finetuning task.

**Questions:**

- How sensitive is performance to tracker noise and missed correspondences. Can you report results with synthetic noise or alternative trackers and include failure cases?
- Have you evaluated on additional multiview or other tasks to demonstrate transfer beyond Probe3D?

---

> ### Author Response · Authors · 2025-11-22
>
> ### **1. Tracking failure cases**
>
> Thank you for requesting clarification on when **3DPoV** may fail. Please refer to the Common response to Adding Failure cases (R-x4UY, R-gkoF)
>
> ---
>
> ### **2. Other benchmarks**
>
> Thanks a lot for this interesting suggestion. We have identified the camera pose estimation evaluation benchmark from VGGT (wang et al., CVPR 2025) and currently working on generating results. We shall update it as soon as possible.
>
> ---
> ### **3. Missing baselines**
>
> We thank the reviewer for bringing these works to our attention. To address the concern regarding comparisons, we searched for their official checkpoints and were only able to locate one for 3DCorrEnhance. Consequently, we compare our method to 3DCorrEnhance on the Probe3D benchmark, as shown in Table 1 and Table 2. From the results, 3DPoV consistently outperforms 3DCorrEnhance across all keypoint-matching metrics, spanning small to large viewpoint changes. Notably, 3DCorrEnhance requires multiview inputs during finetuning, which requires data curation effort, whereas 3DPoV only needs unlabeled videos, an ability that makes it more practical for real-world applications. We will include these new results in the main tables and cite both works, Omniview-Tuning and 3DCorrEnhance, in the related-work section.
>
> _Table 1: SPair-71k Keypoint Matching; Comparison to new baseline, both models are pretrained with DinoV2 with registers_
> |**Model**|**S / 0**|**M / 1**|**L / 2**|**All**|
> |---|---|---|---|---|
> |3DCorrEnhance|59.61|52.16|54.39|54.64|
> |**3DPoV**|**60.16**|**52.79**|**54.50**|**55.40**|
>
> _Table 2: NAVI Keypoint Matching; Comparison to new baseline, both models are pretrained with DinoV2 with registers_
> |**Model**|**$\theta_{0}^{15}$**|**$\theta_{15}^{30}$**|**$\theta_{30}^{60}$**| **$\theta_{60}^{180}$**|
> |---|---|----|---|---|
> |3DCorrEnhance|87.92|67.74|47.18|**31.57**|
> |**3DPoV**|**89.22**|**69.23**|**47.48**|31.33|
>
> ---
>
> ### **4. On the magnitude of improvements**
>
> We appreciate the reviewer’s concern and provide Table 3, which reports the absolute gains of each backbone when finetuned with 3DPoV. We observe improvements of ~1.8% on DINO and ~2% on DINOv2-reg across all viewpoint ranges. The gains on DINOv3 are smaller, largely because this model already exhibits a strong ceiling: it is distilled from a 7B-parameter model trained on large-scale data that is distributionally close to NAVI, making further improvements on this benchmark particularly challenging. Our current results for DINOv3 also use hyperparameters inherited from DINOv2-reg, and we are in the process of optimizing them for the updated model.
>
> To contextualize the magnitude of these gains, Probe3D typically exhibits modest absolute deltas even for state-of-the-art models. For instance, DINOv3 (Simeoni et al. (2025)) reports improvements of +2.6% over DINOv2 (ViT-G) and +1.9% over AM-RADIO (ViT-G) despite using significantly larger architectures and substantially more pretraining compute. In this landscape, 3DPoV’s consistent 1–2% improvements using lightweight video finetuning and no architectural changes are in line with the scale of progress observed on Probe3D, and we believe they represent a meaningful and practical advancement.
>
> _Table 3: SPair-71k Keypoint Matching_
> |**Model**|**Backbone**|**Data**|**S / 0**| **M / 1**|**L / 2**| **All** |
> |---|---|---|---|---|---|---|
> |*Pretrained with DINO*|||||||
> |DINO| ViT-B16|IN-1k|30.19|24.22|24.35|26.39|
> |**3DPoV**|ViT-B16|CO3-YT-DL|**31.77** | **25.74** | **25.80** | **28.16** |
> |*Pretrained with DinoV2-reg*|||||||
> |DinoV2-**reg**|ViT-B14|LVD|58.20|51.56|53.41|53.47|
> |**3DPoV**|ViT-B14|CO3-YT-DL |**60.16**|**52.79**|**54.50**| **55.40** |
> |*Pretrained with DinoV3*|||||||
> |DinoV3| ViT-B16| LVD|61.95|**48.69**|46.77| 55.73|
> |**3DPoV**| ViT-B16 | CO3-YT-DL|**62.24**|48.56|**46.81**|**55.84**|
>
> ### **5. The gains of finetuning on videos**
>
> We ablate the effect of increasing the number of input frames--from a single frame (purely image-based) up to four frames (video-based)--in Table 5(b) of the paper. For convenience, we repeat the results here. The trend is clear: adding more frames consistently improves 3D understanding. The performance gain from 1~$\rightarrow$~2 frames is substantial, while the improvement from 2$\rightarrow$4 frames is smaller.
> Sorting along point trajectories introduces two constraints, boosting features in 3D spatial awareness:
> 1) features belonging to the same trajectory should remain each other’s **first** nearest neighbours, enforcing strong intra-trajectory similarity;
> 2) features should remain distinguishable from features of other trajectories, even when they originate from spatially
> adjacent patches, i.e, the same region, or visually similar objects at different depths.
> These constraints can explain the sharp improvement from 1 to 2 frames. Beyond that, due to the limited variation  in the datasets used, the benefit gradually saturates as more frames are added.

---

> > ### Author Response · Authors · 2025-11-28
> >
> > ## **2. Other benchmarks**
> >
> > To complement Probe3D and assess external validity, we also evaluated our encoder using the V-JEPA v2 (Assran et al., 2025) frozen video-understanding protocol on Something-Something V2(SSv2) (Goyal et al., 2017). While 3DPoV is designed to improve spatial coherence and 3D-awareness, SSv2 contains many samples with inherent depth variation, pose changes and fine-grained object–object interactions. Because spatial reasoning is a key component in interpreting these actions, SSv2 provides a relevant downstream test of whether our spatial improvements translate into better video understanding.
> >
> >
> > The official protocol uses 256×256 inputs and a 16×2×3 sampling pattern (16 frames × 2 temporal crops × 3 spatial crops), trained for 20 epochs. For feasibility, we retained the full validation set and multi-view evaluation but used a reduced configuration: 224×224 resolution, 8×2×3 inputs and a balanced subset of the training data, training the probe for only 5 epochs. Despite this substantially lighter setup, our model achieves higher SSV2 probe accuracy than the DINO baseline (see Table 4), indicating that the spatial regularization introduced by 3DPoV yields more informative representations for downstream action understanding.
> >
> > _Table 4: Frozen SSV2 Video Understanding Performance Using a Linear Probe on Top of the Encoder_
> > |**Model**|**Backbone**|**Acc %**|
> > |---|---|---|
> > |DINOv2R|ViT-B/14|27.73|
> > |**3DPoV**|**DINOv2R-B/14**|**28.66**|

---

> ### Author Response · Authors · 2025-11-28
>
> Dear Reviewer gkoF,
>
> Thank you once again for taking the time and effort to review our submission. As the deadline for the discussion phase is approaching, we wanted to kindly check if you have any additional questions or concerns that we could address in our rebuttal.
>
> We appreciate your insights and look forward to your feedback!
>
> Best regards,
>
> The Authors

---

### Official Review · Reviewer_sdnB · 2025-11-01

**Soundness:** 2
**Presentation:** 2
**Contribution:** 3
**Rating:** 6
**Confidence:** 4

**Summary:**

This paper proposes 3DPOV, a self-supervised, post-training method to improve the 3D spatial understanding and viewpoint invariance of visual foundation models. The core problem it addresses is that models like DINO, while strong in 2D, fail to maintain consistency when camera poses change significantly.

The key idea of 3DPOV is to leverage video to enforce temporal consistency of patch-level feature ordering. Instead of forcing feature descriptors to be identical across time, it trains the model to maintain a consistent relative ranking of patch similarities when compared against a shared set of reference features. The method uses a point tracker (CoTrackerV3) to find patch correspondences across frames, a teacher-student (EMA) framework for stable targets, and differentiable sorting to create a trainable loss objective.

The method is trained on a mixture of video datasets (CO3D, DL3DV, YouTube-VOS) and evaluated on the Probe3D benchmark, where it demonstrates consistent improvements over strong baselines (DINOv2, DINOv3) in keypoint matching, depth estimation, and surface normal estimation, particularly under large viewpoint variations.

**Strengths:**

**Sound Objective:** The core contribution, supervising the relative ordering of patch similarities rather than their absolute values, is a clever learning objective. This relative supervision is probably less sensitive to small appearance variations (e.g., lighting) yet still captures the core geometric/semantic structure, promoting a more consistent similarity space.

**Good Empirical Results:** The method shows clear, consistent gains on the Probe3D benchmark. It critically outperforms baselines (like NeCo) in "medium" and "large" viewpoint shift categories, validating the paper's claim of improving robustness to viewpoint changes.

**Efficiency:** The method is presented as a lightweight post-training (fine-tuning) step, making it a practical way to enhance existing, pre-trained foundation models without the need for full retraining.

**Weaknesses:**

1. Justification of Design Choices: Some key design choices are not well-motivated.

- Inter-Video References: The use of reference patches from other videos in the batch is not well-explained. It's unclear why forcing a patch's similarity ranking relative to irrelevant patches from a different scene would help learn viewpoint-consistent features for the current object. This needs a clearer explanation or ablation.

- Final-Layer-Only Tuning: The paper states that only the final layer is fine-tuned. This is a highly constrained approach for a task that seems to require learning new, fundamental spatio-temporal invariances. It's surprising that this is sufficient, and this choice warrants a stronger justification.

2. Clarity in Presentation: The paper suffers from some minor but important clarity issues.

- Typo in Loss Function: Equation 4 appears to have a typo in its subscripts; the student and teacher permutation matrices are swapped.

- Ambiguous Table Labels: According to L#274 “Unless other stated”, in Table1, 3DPoV under Dino block and 3DPoV under Dino V3 block are same one? Then why there performance is different, or it’s based on different Dino variants?

**Questions:**

Please refer to the questions in the Weaknesses section. If these are addressed satisfactorily, I will consider raising my score.

---

> ### Author Response · Authors · 2025-11-22
>
> ## **1. Justification of Design Choices:**
> -  ### **1.1. Inter-Video References**
>
> The reviewer raises an interesting question. We introduce patches from batch clips (external reference) to ensure diversity in similarity values and scenes. This setup follows the configuration from NeCo. In contrast, the addition of crops from the same clip (internal reference) ensures high-similarity anchors within the broader distribution, sharpening the ranking and ensuring a positive signal for the gradient. Nonetheless, the use of internal crops is limited to the number of frames used in training. We ablated this design choice, reducing the number of references to match the number of frames and observe that indeed exclusive use of internal references result in better performance (+0.32% on 'All'). This suggests that internal reference patches maximize the coverage of the specific dynamic scene, which is more valuable for learning fine-grained 3D correspondence. This observation highlights an area where our method can be further improved, and we will continue to investigate and refine the internal/external reference balance, updating the final version accordingly.
>
> _Table 1: SPair-71k keypoint matching ablations. For a reference pool of size 4 we compare different internal/external splits_
> | **References** | **S / 0** | **M / 1** | **L / 2** | **All** |
> |---|---|---|---|---|
> | External (4)| 59.66| 52.27| 54.40| 54.89|
> | Internal (1) + external (3)| 59.87| 52.44 | 54.27| 55.07|
> | **Internal (4)**| **60.24**| **52.71**| **54.36**| **55.39**|
>
> ---
>
> -  ### **1.2. Final-Layer-Only Tuning**
> We acknowledge the reviewer's concern and wish to clarify that we unfreeze the last two layers and not strictly the final layer. This choice is deliberate and empirically validated, not merely a constraint.
>    - Foundation models e.g. DINOv2 already possess strong implicit 3D awareness (El-Banani et al., 2024). 3DPoV is designed to temporally align these existing spatial features, not to learn 3D geometry from scratch.
>    - Our differentiable sorting loss relies on the pretrained model's local structure to bootstrap learning. Finetuning earlier layers destabilizes these spatial priors, degrading the signal for our permutation loss. We confirmed this empirically (see Table below), where finetuning the last four layers lead to a significant drop in performance. Restricting updates to the last two layers allows the model to encode temporal consistency without forgetting its  spatial representations.
>
> _Table 2: SPair-71k Keypoint Matching; we unfreeze a number of layers and experiment under the same setup_
> |**Unfrozen Blocks**|**S / 0**|**M / 1**|**L / 2**|**All**|
> |---|---|---|---|---|
> |Blocks 8-11 | 57.66| 49.84| 51.48| 52.50|
> |**Blocks 10-11**|**60.16**|**52.79**|**54.50**|**55.40**|
> |Blocks 11|58.64|51.72|53.41|53.79|
>
> ---
> ### **2. Clarity in Presentation**
> Following the reviewers' suggestion, we updated the notation to emphasize the distinctions more clearly in the final version.
>
> Our loss formulation is intentionally designed to strengthen spatial discrimination in patch correspondences. Concretely, we supervise student permutation distributions using a reversed cross-entropy of the form:
> \begin{equation}
> L_{\text{CE}}^{i} = -\sum_{j=1}^{np_r} P_{t,r}^{S}[i,j]\cdot\log\left(P_{0,r}^{T}[i,j] + \epsilon\right),
> \end{equation} where the student distribution acts as the weighing measure.
>
> If we express cross-entropy in terms of KL divergence and entropy,
> \begin{equation}
> CE(P_A,P_B)= KL(P_A\parallel P_B) + H(P_A),
> \end{equation}
> the direction used in prior work such as NeCo, $CE(P_T,P_S)$, reduces (up to constants) to minimizing $KL(P_T \parallel P_S)$ because $H(P_T)$ is fixed when the teacher is frozen (receives EMA updates).
>
> This corresponds to a mode-covering divergence: the student must distribute mass wherever the teacher assigns probability, encouraging broad, soft distributions that cover the teacher's uncertainty.
>
> In contrast, the loss we apply, $CE(P_S,P_T)$, can be expressed as :
> \begin{equation}
> CE(P_S,P_T) = KL(P_S\parallel P_T) + H(P_S),
> \end{equation}
> where $H(P_S)$ is not constant. Minimizing this loss therefore simultaneously reduces $KL(P_S\parallel P_T)$ while suppressing the entropy of the student, promoting high-confidence, sharply peaked ranking distributions.
>
> The optimum of this loss is a deterministic distribution that assigns all mass to the teacher’s highest‑probability candidate, illustrating its mode‑seeking nature. Thus, in ambiguous correspondence cases, our formulation encourages the student to make confident, spatially discriminative predictions rather than reproducing the teacher’s diffuse uncertainty.
>
> _Table 3: SPair-71k Keypoint Matching; ablation on loss direction_
> |**Loss**|**S / 0**|**M / 1**|**L / 2**|**All**|
> |---|---|---|---|---|
> |$CE(t,s)$|58.17|51.29|53.13|53.27|
> |**$CE(s,t)$**|**60.16**|**52.79**|**54.50**|**55.40**|

---

> ### Author Response · Authors · 2025-11-28
>
> Dear Reviewer sdnB,
>
> Thank you once again for taking the time and effort to review our submission. As the deadline for the discussion phase is approaching, we wanted to kindly check if you have any additional questions or concerns that we could address in our rebuttal.
>
> We appreciate your insights and look forward to your feedback!
>
> Best regards,
>
> The Authors

---

### Official Review · Reviewer_x4UY · 2025-11-02

**Soundness:** 3
**Presentation:** 3
**Contribution:** 2
**Rating:** 4
**Confidence:** 3

**Summary:**

The paper proposes 3DPoV, a self-supervised post-training method that improves multiview/3D consistency of vision foundation models by (i) a temporal permutation loss over patch similarities across frames, (ii) a teacher–student scheme with a reference pool for robust supervision, and (iii) a mixture of video datasets (CO3D, DL3DV, YouTube-VOS). The target is to enhance viewpoint-robust features for Probe3D tasks: keypoint matching, depth, and surface normals.

**Strengths:**

1. The teacher–student + differentiable sorting pipeline is clear and easy to integrate with existing ViT backbones. The visibility-weighted loss is a neat, principled way to de-emphasize occluded/out-of-track patches.
2. Consistent improvements across regimes and backbones. The paper claims improvements without trading off small-viewpoint for large-viewpoint performance, and could be helpful for downstream tasks.
3. The method requires minimal computational resources compared to Dinov3 pre-training (20 GPU-hours vs. weeks)

**Weaknesses:**

1. Limited Technical Novelty: While the paper presents a functioning system with promising empirical results, the core technical contribution primarily consists of combining well-established components from existing literature. Specifically, the method integrates: (i) standard point tracking via CoTrackerV3 (Karaev et al., 2024) without modification; (ii) the differentiable sorting mechanism from NeCo (Pariza et al., 2025) and Petersen et al. (2022), simply adapted from spatial to temporal domains; (iii) a conventional teacher-student framework with EMA updates following DINO (Caron et al., 2021); and (iv) temporal consistency objectives previously explored in TimeTuning (Salehi et al., 2023) and MoSiC (Salehi et al., 2024). While this specific combination yields positive results, the paper would be strengthened by either introducing more distinctive technical innovations or providing deeper theoretical insights explaining why this particular configuration enhances 3D understanding beyond empirical observation.
2. Clarity Issue (Minor): Table 2 contains two entries labeled "3DPoV ViT-B16" (rows 6 and 12) without clear differentiation. Based on the table structure and placement after the DINOv3 baseline, the second entry appears to use DINOv3 as the backbone, but this distinction is only clarified in Appendix E rather than the main text. Please consider explicitly labeling these variants in the table itself for clarity.
3. Diminishing Returns on Strong Baselines: More concerning is that when applied to state-of-the-art models, the method shows minimal improvement. Specifically, the gain over DINOv3 in Table 2 (94.40→94.47 on Navi θ₀¹⁵) represents only a 0.07% increase, suggesting the approach provides negligible benefits when applied to modern foundation models.

**Questions:**

Check the weakness above.
1. Why does patch ordering improve surface normal estimation? The connection between similarity rankings and learning surface orientation isn’t clear. Could you elaborate?
2. Which scenes or motions cause 3DPoV to fail? It would be more insightful to include failure cases as well.

---

> ### Author Response · Authors · 2025-11-22
>
> ### **1. Limited technical Novelty**
>
> We thank the reviewer for the comment and wish to clarify that 3DPoV goes beyond a straightforward combination of existing components. While we leverage distillation, point-tracker and differentiable sorting, adapting them to dynamic scenes required specific architectural and supervisory novelties distinct from NeCo. We describe the differences as follows:
> - Unlike NeCo (where only the teacher processes reference frames), **3DPoV** inputs reference frames to the student. This fundamentally shifts the optimization goal: the student must learn features that sort consistently against external scenes. **Table 5(a)** confirms that removing this  causes a significant performance drop, showing that it is not merely a standard distillation setup.
> - We do not use NeCo’s cross-entropy $CE(t,s)$ loss. In dynamic scenes with occlusion, standard CE forces the student to mimic the teacher's uncertainty, leading to diffuse rankings. Instead, we introduce a reverse cross-entropy loss $CE(s,t)$. This encourages the student to be sharper than the teacher while remaining within the teacher's support. Our ablations show that the standard NeCo loss fails in this dynamic setting *(we kindly request the reviewer to refer to our response to R-sdnB where this is addressed in more depth)*.
> - To prove our contribution is not solely on the tracker, **Table 12** compares a 3DPoV using a single frame i.e. with tracker removed against NeCo. Our method maintains consistent performance across difficult viewpoints where NeCo fails. This confirms that our loss formulation and feature-space reference pool provide distinct technical value independent of temporal tracking.
>
> ---
>
> ### **2. Clarity Issue**
>
> We acknowledge the reviewer's comment on ambiguous notation and how it might lead to confusion. We shall update Table 2.
>
> ---
> ### **3. Diminishing Returns on Strong Baselines**
>
> While the absolute gain is modest, it represents a meaningful improvement given the high saturation of DINOv3, which is distilled from a 7B-parameter teacher trained on data distributionally similar to NAVI. Crucially, this improvement is statistically consistent across multiple runs (exhibiting very low standard deviation) and is achieved with orders of magnitude less compute than the original pretraining (our finetuning amounts to only 0.028% of a single DINOv3 epoch - 13 epochs of 9,242 samples using 4 frames comapred to 1689M) demonstrating that 3DPoV can efficiently extract geometric signal. Furthermore, these reported results relied on the DINOv2 hyperparameters; we are currently optimizing the training regime specifically for DINOv3’s architecture and shall to report the findings in the final version.
>
> ---
>
> ### **4. Why does patch ordering improve surface normal estimation?**
>
> When sorting patches based on similarity to a reference view, patches that face the reference camera (frontal) typically yield higher similarity scores than those slanting away. By training the model to correctly rank these similarities, we force it to learn subtle, relative differences in orientation between neighboring patches. This effectively acts as a geometric regularizer i.e. to satisfy the ranking objective, the feature encoder must become sensitive to local surface curvature and perspective distortions, which are the fundamental components of surface normal estimation.
>
> ---
> ### **5. Tracker failure cases**
>
> Please refer to the Common response to Adding Failure cases (R-x4UY, R-gkoF)

---

> ### Author Response · Authors · 2025-11-28
>
> Dear Reviewer x4UY,
>
>
> Thank you once again for taking the time and effort to review our submission. As the deadline for the discussion phase is approaching, we wanted to kindly check if you have any additional questions or concerns that we could address in our rebuttal.
>
> We appreciate your insights and look forward to your feedback!
>
> Best regards,
>
> The Authors

---

### Author Response · Authors · 2025-11-22

### **Common response to Adding Failue cases (R-x4UY, R-gkoF)**

We thank the reviewers for requesting an ablation on failure cases. Our primary failure cases stem from tracker ambiguity in scenes with repetitive textures and occlusion. Since 3DPoV relies on CoTrackerv3 for supervision, it inherits specific limitations. In **Appendix J**,  we observe that in scenes containing multiple similar subjects (e.g., a school of fish), if a tracked subject exits the frame and a visually identical one enters, the tracker may "jump" to the wrong instance. While this creates noisy correspondence labels, 3DPoV is designed to handle such noise via visibility-aware weighting. When tracking becomes unreliable/ambiguous, the predicted visibility score typically drops, automatically down-weighting these samples in the loss to prevent them from corrupting the learned representation.

---

### Meta-Review · Area_Chair_dcj6 · 2026-01-06

**Summary:**

The primary concerns leading to the final decision to reject revolve around the limited technical novelty and marginal empirical gains of the proposed method. Reviewers consistently noted that the core framework appears to be a competent but incremental combination of established components (point tracking, teacher-student distillation, differentiable sorting). While the results demonstrate consistent improvements, the absolute gains are modest, especially on state-of-the-art backbones like DINOv3, raising questions about the practical significance and added value of the approach. Furthermore, the empirical benefits for depth and surface normal estimation were seen as particularly marginal. Although the rebuttal provided additional ablations and clarifications, it did not sufficiently elevate the perceived novelty or impact to overcome these fundamental reservations.

**Reviewer Concerns:**

The rebuttal effectively addressed several specific technical questions. The authors provided additional ablation studies on design choices (e.g., loss formulation, reference pool composition, number of fine-tuned layers), clarified evaluation protocols, and added a new baseline comparison and a downstream task result on SSv2. However, the most critical outstanding concerns remain. The argument that the method's novelty lies in the specific adaptation and combination of existing techniques (reverse cross-entropy, internal reference frames) did not fully convince reviewers who viewed the overall contribution as incremental. The perception of diminishing returns on strong models and the modest magnitude of improvements across key tasks were not decisively overturned by the new data.

**Reviewer Scores:**

No reviewers responded during the rebuttal period.

---

### Decision · Program_Chairs · 2026-01-26

Reject